



# Snow accumulation over the world's glaciers (1981-2021) inferred from climate reanalyses and machine learning

Matteo Guidicelli[1], Matthias Huss[1,2,3], Marco Gabella[4], and Nadine Salzmann[5,6]

[1]Department of Geosciences, University of Fribourg, Fribourg, Switzerland
[2]Laboratory of Hydraulics, Hydrology and Glaciology (VAW), ETH Zurich, Zurich, Switzerland
[3]Swiss Federal Institute for Forest, Snow and Landscape Research (WSL), Birmensdorf, Switzerland
[4]Federal Office of Meteorology and Climatology MeteoSwiss, Locarno-Monti, Switzerland
[5]WSL Institute for Snow and Avalanche Research SLF, Davos, Switzerland
[6]Climate Change, Extremes and Natural Hazards in Alpine Regions Research Center CERC, Davos, Switzerland

**Correspondence:** Matteo Guidicelli (matteo.guidicelli@unifr.ch)

**Abstract.** Although reanalysis products for remote high-mountain regions provide estimates of snow precipitation, this data is inherently uncertain and assessing a potential bias is difficult due to the scarcity of observations, thus also limiting their reliability to evaluate long-term effects of climate change. Here, we compare the winter mass balance of 95 glaciers distributed over the Alps, Western Canada, Central Asia and Scandinavia, with the total precipitation from the ERA-5 and the MERRA-2 reanalysis products during the snow accumulation seasons from 1981 until today. We propose a machine learning model to adjust the precipitation of reanalysis products to the elevation of the glaciers, thus deriving snow water equivalent (SWE) estimates over glaciers uncovered by ground observations and/or filling observational gaps. We use a gradient boosting regressor (GBR), which combines several meteorological variables from the reanalyses (e.g. air temperature, relative humidity) with topographical parameters. These GBR-derived estimates are evaluated against the winter mass balance data by means of a leave-one-glacier-out cross-validation (site-independent GBR) and a leave-one-season-out cross-validation (season-independent GBR). Both site-independent and season-independent GBRs allowed reducing (increasing) the bias (correlation) between the precipitation of the original reanalyses and the winter mass balance data of the glaciers. Finally, the GBR models are used to derive SWE trends on glaciers between 1981 and 2021. The resulting trends are more pronounced than those obtained from the total precipitation of the original reanalyses. On a regional scale, significant 41-year SWE trends over glaciers are observed in the Alps (MERRA-2 season-independent GBR: +0.4 %/year) and in Western Canada (ERA-5 season-independent GBR: +0.2 %/year), while significant positive/negative trends are observed in all the regions for single glaciers or specific elevations. Negative (positive) SWE trends are typically observed at lower (higher) elevations, where the impact of rising temperatures is more (less) dominant.

## 1  Introduction

Climate change considerably alters the high-mountain cryosphere, including importantly the seasonal snow regime (e.g. Bormann et al., 2018), which is leading for instance to reduced water storage capacity in worldwide high-mountain regions and



adjacent lowlands (e.g. Viviroli et al., 2007; Immerzeel et al., 2020). With rising air temperatures, snow accumulation is starting later in autumn and melt is occurring earlier in spring (e.g. Beniston, 2012; Vorkauf et al., 2021). Simultaneously, the zero-degree isotherm is rising and more precipitation is already falling as rain rather than snow, particularly at lower elevations

(e.g. Marty, 2008; Beniston et al., 2018). Thus, the balance between temperature and precipitation changes will determine whether snow accumulation will increase or decrease (Stranden and Skaugen, 2009; Sospedra-Alfonso et al., 2015). Increased winter precipitation will only have a positive impact on the snowpack at very high elevations (e.g. Marty et al., 2017), where the increase of precipitation dominates over the increase of temperature. However, the elevation dependency of precipitation trends is unclear: precipitation trends in station observations are often inconsistent with no systematic changes with elevation,

while precipitation increases in gridded datasets are weaker at higher elevations (e.g. Pepin et al., 2022).

In addition to water shortage, cryospheric hazards such as slope failures and glacier lake outburst floods (e.g. Gobiet et al., 2014; Rasul and Molden, 2019) are other impacts of climate change in mountain regions, which are typically felt by society at the local scale (e.g. Adger et al., 2003; Hock et al.). It is thus crucial to improve our understanding of the local climate-cryosphere interaction in order to implement appropriate adaptation strategies (e.g. Stone et al., 2013; Salzmann et al., 2014;

Huss et al., 2017; Barandun et al., 2020). However, at the local scale and particularly at very high altitudes, snow (and precipitation) data is typically very scarce, spatially not optimally distributed, with low temporal resolution or time series are short or with important gaps because of technical challenges, difficult accessibility and thus complicated and lavish maintenance (e.g. Beniston et al., 2012; Tapiador et al., 2012). This is an important limitation for studies focusing on the long-term effects of climate change on the snowpack which require snow accumulation data covering decadal periods (e.g. Seiz et al., 2010). Thus,

the further development of techniques to spatially and/or temporally transfer the available observational series between sites and/or filling data gaps, is critical and urgently needed (e.g. Salzmann et al., 2014).

Understanding of snow accumulation dynamics, as well as total precipitation at the highest parts of mountain ranges is crucial, as – despite their relatively limited areal extent – these regions act as water towers, releasing vital amounts of water in the form of glacier and snow melt in the dry summer season, i.e. when it is most urgently needed (e.g. Immerzeel et al., 2020).

Due to the scarcity and inherent problems of conventional precipitation gauges in high-mountain environments dominated by snow and strong winds (e.g. Sevruk et al., 2009), snow water equivalent (SWE) at very high elevations is mostly measured as the cumulative snow precipitation on glaciers over the accumulation season. The snow temporally stored by glaciers during the accumulation season is a critical component of the high-mountain cryosphere (e.g. Barnett et al., 2005). As melting is often negligible during this time period, SWE on glaciers represents a reliable measure of local winter precipitation and was

thus used for a comparison with precipitation products in different studies (e.g. Gugerli et al., 2020; Guidicelli et al., 2021). Measurements of SWE on glaciers are typically used for the determination of winter mass balance (Cogley et al., 2011), an important variable in international glacier monitoring (e.g. Zemp et al., 2013). The main process of snow accumulation is the total precipitation received by the glacier during the accumulation season (mainly snowfall but including also deposition of hoar and freezing rain). However, other processes as snow drift caused by winds and avalanching can also influence the

accumulation (Dadic et al., 2010; Gascoin et al., 2013). Currently, the correct partitioning of mass fluxes on glaciers puts limits on the usefulness of many large-scale remote sensing data sets: whereas, the volume change of glaciers over time can be inferred





with remote sensing observations (e.g. Hugonnet et al., 2021), the unambiguous calibration of glaciological and hydrological models requires the two components of glacier mass balance (accumulation and melt), which cannot be constrained with remote observations. The improvement of snow accumulation estimates in observation-scarce regions is thus highly relevant.

Worldwide spatio-temporally continuous information on precipitation, snow depth and SWE is also provided by climate reanalyses that merge physical laws with the assimilated satellite and ground observations (e.g. Hersbach et al., 2020; Gelaro et al., 2017). However, the performance of reanalysis results can vary greatly depending on the region and the elevation range of interest (Sun et al., 2018). The scarcity of observations that can be assimilated and the coarse resolution of such models limit their accuracy in areas of complex topography and their suitability for studies at a local scale (e.g. Salzmann and Mearns, 
2012, (for snow)).

In fact, the downscaling of precipitation estimates of reanalyses is necessary to represent the local conditions in high-mountain regions. Different downscaling methods exist (cf. Maraun et al., 2010). For instance, Liston and Elder (2006) developed a quasi–physically based, meteorological model to produce high-resolution (30 m to 1 km horizontal grid) atmospheric forcings for several variables, where the precipitation adjustment is a nonlinear function of the elevation difference between 
the grid and the point of interest. The same equation was used by Gupta and Tarboton (2016), who proposed an approach to downscale the MERRA (Modern-Era Retrospective analysis for Research and Applications) variables. They obtained a Nash-Sutcliffe efficiency greater than 0.70 for downscaled monthly precipitation at 173 SNOTEL (Snow Telemetry) sites. Fiddes and Gruber (2014) adapted this method for the Swiss Alps by including a climatological parameter based on the Alpine precipitation data set provided by the Climatic Research Unit (gridded monthly precipitation totals at 10 arc-min resolution over the Alps, 
for the period 1800–2003). Their product allowed improving the purely lapse-rate-based approach of Liston and Elder (2006), obtaining a correlation coefficient of 0.6 (versus 0.5) against the yearly precipitation observed at 40 ANETZ (MeteoSwiss automatic meteorological network) stations. Recently, machine learning methods have demonstrated their high performance to statistically downscale reanalyses (and global climate models) estimates of precipitation and other meteorological variables, from sub-daily and daily (e.g. Serifi et al., 2021; Wang et al., 2021) to monthly and seasonal (e.g. Sachindra et al., 2018; Najafi 
et al., 2011; Sun and Tang, 2020) resolution. However, downscaling methods for snow (and precipitation) are rarely assessed at very high elevations, to a great part because of the scarcity of ground observations. Consequently, long-term effects of climate change on the snowpack at very high elevations are not well understood yet (e.g. Seiz et al., 2010).

In this study, we thus aim at providing improved observation-independent SWE estimates and exploiting the results to infer the relation of precipitation and temperature changes with SWE trends at the highest elevations of different mountain ranges 
across the Earth. In order to achieve this goal, we develop and evaluate a machine learning approach based on gradient boosting regressor (GBR) models to adjust the total precipitation of reanalysis dataset (ERA-5 and MERRA-2) over the accumulation season. The GBR model is then used to derive SWE estimates over glaciers in the Alps, Scandinavia, Central Asia and Western Canada. Data on snow accumulation distribution at the end of the accumulation season covering a period of up to 41 years from 95 glaciers were used to train our approach. More specifically, the GBR models aim at allowing the spatio-temporal 
transferability of the learned information over the 95 glaciers to other glaciers with no ground observations and/or filling gaps





of observational series. The new information provided by our study can be very helpful to further evaluate the local impact of climate change on snow over glaciers in different regions of the world.

## 2 Study sites and data

The study was conducted on 95 glaciers located in the Alps, Scandinavia, Central Asia and Western Canada (Fig. 1), where the
longest time series and the highest density of winter glacier mass balance data are available. In the following, we describe the different data sources used in the study.

### 2.1 Reanalysis data

We used data from ERA-5 and MERRA-2 reanalyses since they are among the currently most used reanalysis products with the highest spatial resolution, covering the longest time period in all the regions of our study.

### 2.1.1 ERA-5

ERA-5 is the fifth generation of the European Centre for Medium-Range Weather Forecasts atmospheric reanalyses of the global climate (see Hersbach et al., 2020, for more information). In this study, we used several variables from the ERA5 hourly data on single levels from 1979 to present (Hersbach et al., 2018b), and the ERA5 hourly data on pressure levels from 1979 to present (Hersbach et al., 2018a), all with a spatial resolution of 0.25° x 0.25° (∼30 km). All variables were resampled on a
daily timescale before usage. The list of variables selected for the analysis is reported in Table B1. The ERA-5 precipitation variable used in the study is "tp" (total precipitation) from the ERA-5 single levels.

### 2.1.2 MERRA-2

MERRA-2 is the second version of the Modern-Era Retrospective Analysis for Research and Applications (see Gelaro et al., 2017, for more information). In this study, we used several variables from the MERRA-2 Land Surface Diagnostics (Global
Modeling and Assimilation Office (GMAO), 2015b), the MERRA-2 Single-Level Diagnostics (Global Modeling and Assimilation Office (GMAO), 2015c) and the MERRA-2 Analyzed Meteorological Fields (Global Modeling and Assimilation Office (GMAO), 2015a). All variables have a spatial resolution of 0.5° x 0.625° (∼50 km), and we resampled them on a daily timescale before usage. The list of the selected variables is reported in Table B2. The MERRA-2 precipitation variable used in the study is "PRECTOTLAND" (total precipitation) from the MERRA-2 Land Surface Diagnostics.

### 2.2 Winter mass balance data

The World Glaicer Monitoring Service (WGMS) compiles and publishes standardized observations on changes in mass, volume, length and area of glaciers collected by national monitoring programmes and local observers around the world (glacier fluctuations (see Zemp et al., 2021, for more details)).





We used the winter mass balance data separated per elevation intervals (EE-MASS-BALANCE data sheet in WGMS, 2020)
and we refer to them as $B_w$ in this study. Point observations are also available (EEE-MASS-BALANCE POINT data sheet in
WGMS, 2020) but are not used in this study because of the smaller amount of glaciers with complete information reported
(observation dates, elevation, coordinates). We only considered the $B_w$ data where the elevation interval is indicated in the
WGMS database. The glacier area related to each elevation interval was also used to weight the $B_w$ data. In addition, we
considered the average slope and aspect of the glaciers by using the information provided in the Randolph Glacier Inventory
version 6 (RGI Consortium, 2017).

The winter mass balance is the result of the balance between the gain of snow which accumulates over the glacier, as well as
refreezing of liquid water within the snowpack, and the loss caused by melting and sublimation over the accumulation season.
Other processes such as snow drift caused by winds can also influence the accumulation. However, the amount of snow melt
is typically minor compared to the snow accumulation and is thus neglected in the comparison with the precipitation totals
performed in this study. Snow accumulation is expressed in SWE (e.g. Østrem and Brugman, 1991), which is calculated by
multiplying the measured snow depth with the respective bulk density of the snowpack. The snow depth is typically measured
with a snow probe or ground-penetrating radar, while the snow density is usually measured in snow pits or by coring and is
subsequently extrapolated to all observations on a glacier. The WGMS database only provides information on SWE but does
not generally allow tracing whether density was directly measured or not.

The $B_w$ data used in this study correspond to the mean winter balance for the glacier area contained in the respective ele-
vation interval. Various spatial extrapolation techniques were applied by the national observers to infer elevation band average
snow accumulation from the (sparse) point observations, which can be challenging due to important local-scale variability in
snow depth (e.g. Dadic et al., 2010; Helfricht et al., 2014; Sold et al., 2016). Unfortunately, the WGMS database does not
allow tracing the methods used, hence, resulting in an uncertainty that is difficult to be estimated. Often, no direct snow depth
and density observations are available at the most extreme elevations of the glaciers because of high surface slopes and diffi-
cult accessibility. The employed techniques in the framework of the Swiss national program GLAMOS (Glacier Monitoring
Switzerland, providing the data for the Swiss sites to the WGMS) are described in Huss et al. (2021). The impact of the inter-
and extrapolation of direct SWE measurements acquired on glaciers to obtain $B_w$ data used in this study on our results is
discussed in Section 5.1.3.

The starting date of the accumulation season is not precisely known and is often determined with a stratigraphic system
(since the date of the minimum surface in the previous summer) (e.g. Mayo et al., 1972; Cogley et al., 2011). The date of the
minimum surface varies between the years and also across the glacier, in fact, the snow accumulation starts typically later at
lower elevations than at higher elevations (Huss et al., 2009). However, in this study we used a unique starting date for the entire
glacier according to the information provided in the WGMS database. The end of the season is determined by the day of the
snow survey that is indicated in the WGMS database. In this study, we cumulated precipitation amounts over the accumulation
season. The impact of the date considered as beginning of the accumulation season on our results is discussed in Section 5.1.3.





## 3 Methods

First, we derived total or average of all variables provided by the reanalyses for the entire accumulation season. Subsequently, a machine learning model to adjust the total precipitation of the reanalyses over glaciers for the accumulation season was devel-
oped to derive SWE estimates. We use a GBR, which makes use of several meteorological variables (original and downscaled) and topographical parameters as input variables (predictors). The applied methods to downscale other meteorological variables used by the GBR model are described below.

### 3.1 Downscaling temperature and relative humidity

In addition to the original variables, the GBR requires some downscaled variables of the reanalyses as predictors at the glacier
elevation, including air temperature, dew point temperature and relative humidity (for MERRA-2 and ERA-5), vertical velocity of air motion (for ERA-5 only) and specific humidity (for MERRA-2 only). The downscaling procedure was applied at a daily resolution using a linear interpolation between the values of the two closest pressure levels to the elevation of the $B_w$ data of the glaciers. The downscaling approach is illustrated in Fig. 2.

If information regarding the relative humidity was not directly provided by the reanalyses, we applied approaches presented
by Liston and Elder (2006) and Gupta and Tarboton (2016) to derive it. The applied equations are described in the Appendix (Section A).

### 3.2 Downscaling precipitation

In order to derive downscaled precipitation estimates over the glaciers, we built a machine learning model and we applied a pre-existing lapse-rate-based approach that we considered as a benchmark. The approaches are described below and the results
obtained with the two methods are compared afterward.

#### 3.2.1 Benchmark

Liston and Elder (2006) proposed a lapse-rate-based approach to downscale reanalysis' precipitation by accounting for the elevation difference between the point of interest and the grid of the reanalysis. Whereas they applied the approach to the MERRA reanalysis, we applied the same approach to MERRA-2 and ERA-5 reanalysis data and used the resulting adjusted
precipitation as a benchmark:

$$P_{\mathrm{adj}} = P_{\mathrm{reanalysis}} \frac{1 + \kappa(H_{\mathrm{point}} - H_{\mathrm{reanalysis}})}{1 - \kappa(H_{\mathrm{point}} - H_{\mathrm{reanalysis}})}, \tag{1}$$

where $P_{\mathrm{reanalysis}}$ is the precipitation of the reanalysis, $H_{\mathrm{reanalysis}}$ is the elevation of the reanalysis' grid, $P_{\mathrm{adj}}$ is the adjusted precipitation at the altitude of the point of interest ($H_{\mathrm{point}}$) and $\kappa$ is a monthly adjustment factor (cf. table 1 of Liston and Elder, 2006). In our study, we used an average factor $\kappa = 0.3214$, corresponding to the average between October and April.
The precipitation adjusted with this approach on Findelgletscher is illustrated in Fig. 2.





### 3.2.2 GBR model (Gradient Boosting Regressor)

In order to represent a potential non-monotonic increase of snow accumulation (and precipitation) with elevation and to provide different adjustments of the original reanalysis' precipitation depending on the region and the site, we built more complex models based on machine learning. All the models are built with the open source "scikit-learn" library for machine learning in Python (cf. Pedregosa et al., 2011). Specifically, we built a series of GBR models, each one consisting of an ensemble of weak learning models (estimators) represented by regression trees. In our case, the goal of the GBR models is to predict the logarithmic adjustment factors of the reanalysis' precipitation with respect to the $B_w$ on glaciers (Eq. 2). In a GBR, the trees are built sequentially, and the subsequent trees learn from the errors of the previous trees, minimizing the residuals between their predictions and the reference values (cf. Fig. 3).

$$F_{\mathrm{dB,ref}} = 10 \log_{10} \frac{B_w}{P_{\mathrm{reanalysis,tot}}}, \tag{2}$$

where $P_{\mathrm{reanalysis,tot}}$ is the total precipitation of the reanalysis over the accumulation season. The GBR models aim at minimizing the cost function defined in Equation 3, corresponding to the mean squared error between the predicted and reference logarithmic adjustment factors.

$$MSE_{\mathrm{dB^2}} = \frac{1}{n} \sum_{i=1}^{n} (F_{\mathrm{dB,pred},i} - F_{\mathrm{dB,ref},i})^2 \tag{3}$$

Different hyperparameters characterize a GBR. In this study, we applied a grid search to optimize the number of estimators (number of additive trees), the maximum depth that each tree can reach, the minimum number of samples required to be at a leaf node of a tree, and the maximum number of predictors that are randomly selected at each split for each tree (the predictor reducing the error the most is used to split the node). The hyperparameters that were able to minimize the mean squared error of the validation data were chosen. The optimal values are reported in Table 1.

The validation data was defined differently depending on the goal of the GBR model. For both reanalysis products (ERA-5 and MERRA-2), we built two different GBR models with two different goals and two different cross-validation schemes (see Fig. 3). The first one is site-independent and aims at "extrapolating" the $B_w$ data in time and space (over glaciers with no $B_w$ data). We thus applied a leave-one-glacier-out cross-validation. The second one is season-independent and aims at "extrapolating" the $B_w$ data in time only (filling data gaps over glaciers with discontinuous records of $B_w$). For these cases, we applied a leave-one-season-out cross-validation. Thus, the site-independent model is more generalized (no information regarding the glacier where the model is validated was provided), while the season-independent model is more detailed and performs better over glaciers with $B_w$ data but worse over glaciers with no $B_w$ data.

All the variables presented in Section 2.1 and listed in Section B of the Appendix were used by the GBR as predictors (separately for ERA-5 and MERRA-2). In addition, we derived and used the differences between the downscaled variables (cf. Section 3.1) and the estimates at the grid of the reanalysis. Some variables were not only averaged considering all days in the accumulation season, but also considering only the days with a relevant amount of precipitation, here arbitrarily set to 5 mm. The GBRs also use the latitude and longitude of the glacier (unique value for the entire glacier), as well as the aspect





and slope of the glacier (unique value for the entire glacier). A summary of the predictors used by the GBRs is reported in Table B3. Furthermore, the $B_w$ data were weighted by considering the area of the glacier related to the respective elevation

interval. Larger glaciers (and elevation intervals related to larger areas) thus receive more weight in the training of our models than smaller glaciers (and elevation intervals related to smaller areas).

## 3.3 Evaluation metrics for the models' estimates

### 3.3.1 Adjustment factors

In order to evaluate the bias between the $B_w$ data and the estimates of the models (original reanalyses, benchmark or GBR),

we computed the adjustment factor $f$ (dimensionless) as:

$$f = \frac{B_w}{E_{\text{model}}},\qquad(4)$$

where $E_{\text{model}}$ is the estimate of a model. The adjustment factor $F_{\text{dB}}$ is expressed in decibels and is used to derive supplementary evaluation metrics:

$$F_{\text{dB}} = 10\log_{10}\frac{B_w}{E_{\text{model}}}\qquad(5)$$

### 3.3.2 Glacier-wide means

When deriving a glacier-wide factor (or glacier-wide $B_w$) for a single accumulation season, we computed a weighted mean of the area contained in the individual elevation intervals. These seasonal glacier-wide values were used to derive the Pearson's correlations (CORR), the root mean square error (RMSE), and the fraction of standard error (FSE), between the glacier-wide $B_w$ and the model estimate. The FSE corresponds to the RMSE divided by the glacier-wide $B_w$.

### 3.3.3 Regional metrics

In order to further validate the performance of the GBR models, we derived the glacier-wide $F_{\text{dB}}$ described by Equation 5 for every accumulation season and every glacier with $B_w$ data ($F_{\text{dB, mean}}$). We thus analyzed the four investigated regions separately by deriving a mean factor per region, as:

$$F_{\text{dB, region}} = \frac{\sum_{g=1}^{n}\sum_{s=1}^{m_g}F_{\text{dB,mean},g,s}}{n\sum_{g=1}^{n}m_g},\qquad(6)$$

where $n$ is the number of glaciers and $m_g$ is the number of accumulation seasons with $B_w$ data for the glacier $g$.

## 3.4 Computation of climatological trends

A climatological analysis of the precipitation and SWE trends over the glaciers was made by using, exclusively, the precipitation estimates of the original reanalyses and the SWE estimates of the GBR models. For every year between 1981 and 2021, we considered a fixed time period from the first of October to the 30th of April. Furthermore, we considered fixed elevation





intervals of 100 m between the minimum and maximum elevation of each glacier. Therefore, each glacier was attributed 41
      estimates (41 accumulation seasons) for each elevation interval covered. In order to determine the significance (p-values) of
      the trends, we applied the Mann-Kendall trend test (Mann, 1945; Kendall, 1975), which was often applied to assess hydro-
      meteorological trends (e.g. Ahmad et al., 2015; Wang et al., 2020) and is recommended by the World Meteorological Organi-
      zation for studying climatic series (Hisdal et al., 2001). Thus, we defined the absolute trend as the non-parametric Sen's slope

estimator, representing the median of all the slopes determined by the pairs of points (Sen, 1968). Finally, we derived relative
      trends in order to compare regions with large differences in terms of overall precipitation. For each glacier, the relative SWE
      trend is thus calculated by dividing the Sen's slope estimator by the mean of the GBR model estimates of the glacier for the
      period 1981-2021.

## 4   Results

In the following, we firstly report the analysis of the predictors' importance in the GBR models (Sec. 4.1). Then, we present
      the main results of our study, i.e. the performance of the GBR models (Sec. 4.2) and its subsequent use for the derivation of
      multi-decadal SWE trends over glaciers in the Alps, Scandinavia, Central Asia and Western Canada (Sec. 4.3).

### 4.1   Importance of predictors in the GBR models

In order to understand the importance of the predictors used by the GBR models, we evaluated the number of times each
predictor was used to split a node in the regression trees. For both GBR models, the most frequently used predictor is the
      elevation difference between the $B_w$ data and the underlying terrain model of reanalysis, followed by other predictors which
      are also related to the elevation difference (downscaled temperature, relative humidity, etc.) (Fig. 4a and b). Among the other
      predictors, we notice that surface pressure, latitude/longitude, slope, aspect and the predictors characterizing the topographical
      complexity of the reanalysis' grid cell are also frequently used in both GBR models. We furthermore performed a principal
component analysis (PCA) considering the ten predictors most frequently used by the GBRs. For both ERA-5 and MERRA-2,
      clusters of points related to the individual regions are recognized (Fig. 4c and d). For the PCA based on MERRA-2 (Fig. 4d),
      Scandinavian sites, located at lower elevation, have generally lower factors between $B_w$ and MERRA-2 precipitation than the
      sites located in Western Canada, Central Asia and in the Alps.

In order to infer the importance of the predictors for the individual regions, we also built a different GBR for each region
separately. The ten most used predictors in the GBRs were selected, and we performed again a PCA for each region separately.
      In the Alps, lower factors between $B_w$ and ERA-5 precipitation are observed at lower latitudes, and the glaciers affected by
      100 m westerly winds (negative $u$ component of the wind speed) seem to have generally higher factors than those affected
      by easterly winds (Fig. 5a). In Scandinavia, we notice a cluster of five glaciers with smaller ERA-5 factors and warmer
      downscaled temperatures during precipitation events (Fig. 5c). In Central Asia, the glaciers' aspect is the predictor that most
clearly discriminates between high and low factors between $B_w$ and both ERA-5 and MERRA-2 precipitation. Glaciers with
      ∼North-facing slopes have smaller ERA-5 factors and ∼East-facing slopes have higher MERRA-2 factors (Fig. 5e and f). In





Western Canada, lower ERA-5 factors can be explained by the larger precipitation and lower elevation of the glaciers, while MERRA-2 factors are clearly lower at higher latitudes, which are characterized by stronger southerly winds at 850 hPa (Fig. 5g and f).

## 4.2 Performance of the GBR models

Overall, the GBR models indicate better agreement in terms of bias, spatial and temporal correlation with the $B_w$ data than the original reanalyses and the benchmark for the majority of the studied glaciers. In the following we report in detail on the comparison between the $B_w$ data, the precipitation estimates of the reanalyses and the GBR models' estimates.

### 4.2.1 Glacier-wide reanalysis' bias adjustment

Figure 6 shows the comparison between all glacier-wide $B_w$ values and the models' estimates. MERRA-2 precipitation underestimates $B_w$ more importantly than ERA-5 precipitation in all regions (Fig. 6a and b), with an overall RMSE of 946 mm against 793 mm of ERA-5. Excluding the Alps, the correlation between the $B_w$ data and the ERA-5 precipitation is always higher than the correlation with the MERRA-2 precipitation. The adjusted estimates obtained with the site-independent and the season-independent GBRs allowed us to consistently reduce (increase) the bias (correlation) between the precipitation of the 285 original reanalyses and $B_w$ (from an overall RMSE (CORR) of 946 mm (0.74) and 793 mm (0.81) of MERRA-2 and ERA-5, to 443 mm (0.85) and 422 mm (0.86) of the site-independent GBRs, and 287 mm (0.94) and 272 mm (0.95) of the season-independent GBRs). These results demonstrate the need of an adjustment of reanalyses data to reproduce snow accumulation on glaciers, which are, otherwise, largely underestimated in all four regions involved in this study.

In order to make an in depth analysis of the model performance, we also derived a glacier-wide factor between the $B_w$
data and reanalysis-based models' estimates (Eq. 4) for each accumulation season and for each site separately (Fig. 7). By comparing Fig. 7b and 7c, it is clear that, in Central Asia, the factors for adjusting the MERRA-2 reanalysis' precipitation are much larger than the factors for the ERA-5 precipitation. The benchmark method overestimates the $B_w$ for many glaciers in the Alps (both ERA-5 and MERRA-2) and several glaciers in Central Asia (ERA-5). The site-independent and especially, the season-independent GBRs are better scaled with respect to the $B_w$ data than the original reanalyses and the benchmarks. In 295 general, the variability of the factors for each glacier is strongly affected by the number of available accumulation seasons with $B_w$ data (Fig. 7a). A lower variability is usually observed for glaciers with a small number of seasons with $B_w$ data.

Figure 8 shows the mean regional factor between the $B_w$ data and the models' estimates as a function of the accumulation seasons from 1981 to 2019. It indicates that the original reanalyses clearly underestimate snow accumulation on glaciers, except for ERA-5 in Central Asia, where, as a consequence, the benchmark overestimates $B_w$. However, temporal variations 300 in the mean regional bias are also affected by the considered set of glaciers that fluctuates over the analyzed years. In the Alps, we observe increasing biases of the original reanalyses in recent years, where a much larger number of glaciers is available. In Scandinavia, the bias of MERRA-2 and ERA-5 is similar and all the models are generally not able to remove it completely. In, Central Asia, there is a tendency for all models to yield lower adjustment factors before the 2000s than afterwards. However,





this has to be interpreted with care, because only one glacier was considered between 2002 and 2014. The continuity of the

available $B_w$ data in Western Canada is too limited to analyze temporal changes in the adjustment factors.

In order to evaluate the robustness of the GBR models to reduce the glacier-wide bias of the reanalysis, we performed a temporal and spatial validation of their predictions (Fig. 9). The performance of the season-independent models improves when using more accumulation seasons in the training data (Fig. 9a, c, e and g). Training the models with more than 20 seasons, however, does not seem to further improve performance consistently. The performance of the site-independent models

is constant because they are never trained with $B_w$ data of the validated glacier (in the cross-validation). When no $B_w$ data of the validated glacier is used to train the season-independent models (as for the site-independent models), their performance is worse than the site-independent models, confirming the importance of a specific optimization scheme depending on the goal of the model.

As also expected, the performance of the site-independent models decreases when data of neighbouring glaciers are excluded

from the training (Figs. 9b, d, f and h). The highest impact is on the performance of the MERRA-2 site-independent GBR in Central Asia. Overall, the bias of the site-independent GBR models remains comparable to the bias obtained with the benchmark method even when excluding all other glaciers located within 1000 km from the training. For the season-independent models, we always kept the $B_w$ data of the validated glacier in the training data, and only excluded the other glaciers. This explains why the season-independent models perform better and are less sensitive to the removal of neighbouring glaciers from

the training process.

The good performance of the GBRs in terms of bias suggests that they can be used for SWE estimates over glaciers where no ground observations are available (site-independent GBRs) and for filling data gaps of the recorded observations (season-independent GBRs). The results indicate moreover that the season-independent GBRs outperform the site-independent GBRs to reduce the bias against $B_w$ data, especially in regions with a limited number of glaciers with snow accumulation data. In

conclusion, filling data gaps is much simpler than estimating SWE on glaciers with no observations.

### 4.2.2  Spatial snow accumulation variability on single glaciers

In order to evaluate the ability of the GBR models to reproduce the spatial variability of the snow accumulation over single glaciers, we compared the vertical profiles of $B_w$ to the estimates of the models. For Rhonegletscher for instance (Alps, Fig. 10a), both site-independent and season-independent GBRs are able to represent the shape of the vertical profile of the $B_w$,

which is characterized by an increasing $B_w$ until 3350 m a.s.l. and a more stable/decreasing $B_w$ in the upper part of the glacier. This vertical profile cannot be reproduced by using the benchmark approach, where, by definition, the precipitation is monotonically increasing with the elevation. Decreasing $B_w$ is also clearly indicated in the upper part of Abramov glacier (Central Asia, Fig. 10b) in 1992. As suggested by the point observations reported, this is certainly the result of extrapolating to elevation ranges not or only poorly covered with data. However, this has a limited influence on the GBR models than the lower

part of the glacier, as it received higher weights because of the larger areas (see Sec. 2.2). The total precipitation estimated by the original reanalyses was well reproduced compared with the $B_w$. The site-independent GBRs are not able to adjust the precipitation by consistently reducing the bias with $B_w$. On the other hand, the season-independent GBRs are able to better



fit the altitudinal distribution of $B_w$. In this case, we observe that the maximum $B_w$ coincides with the maximum downscaled MERRA-2 relative humidity. In the case of Storglaciären (Scandinavia, Fig. 10c), $B_w$ is underestimated by the benchmark,

while the GBR models (the season-independent especially) are able to represent the steep increase of snow accumulation over the glacier. In the case of Sykora glacier (Western Canada, Fig. 10d), all GBR models show a good agreement with $B_w$ data. By comparing the coefficients of variation, it is clear that the season-independent GBRs are able to reproduce better the amplitude of the spatial variability of the $B_w$ than the site-independent GBRs (Table 2). Furthermore, the correlations demonstrate that the GBRs outperform the benchmark method to reproduce the $B_w$ of almost all glaciers of this study (Table 2).

### 345  4.2.3  Temporal snow accumulation variability on single glaciers

The GBR models have also generally shown a better ability to reproduce the relative changes of $B_w$ among individual years for the same glacier than the original reanalysis (Tab. 3). In fact, the correlation between the GBR models' estimates and the $B_w$ over the years is often much higher than for the original reanalysis. The level of significance of the correlation between the original ERA-5 or/and MERRA-2 improves when the GBR models are applied on Adlergletscher, Findelgletscher, Vedretta

Pendente, Goldbergkees, Breidablikkbreen, Graasubreen, Abramov glacier and Ts.Tuyuksuyskiy glacier. However, in some cases low correlations are still observed and the level of significance decreases when the ERA-5 season-independent approach is applied (Ts. Tuyuksuyskiy glacier), or the MERRA-2 season-independent approach is applied (Ghiacciaio del Basodino, Ghiacciaio del Ciardoney, Kleinfleisskees), indicating that the models are not suitable to represent the temporal variability in these cases. Furthermore, although the season-independent GBRs are the best models to reduce the bias, the relative changes

among years are sometimes better explained by the site-independent GBRs. We thus monitored the correlation with the available $B_w$ data over the years to evaluate the reliability of the SWE trends derived with the GBR models in the following section. Of course, the number of years with $B_w$ data strongly influences the correlation and its significance, which is typically lower in Western Canada and Central Asia, where the available number of years per glacier with $B_w$ data is smaller than in the Alps and Scandinavia (see Fig. 7a).

Overall, the GBR models have shown a higher performance than the original reanalyses in terms of bias and temporal correlation with snow accumulation data on glaciers. We thus suggest that our new estimates can be used to derive SWE trends, providing an important feedback of the relation between climate change and both snow accumulation and precipitation at the highest elevations of mountain ranges where often no direct precipitation records are available.

### 4.3  SWE trends (1981-2021)

We used our GBR models trained with $B_w$ on glaciers in order to derive SWE estimates over glaciers on the 30th of April (i.e. close to the end of the accumulation season) and derive related trends between 1981 and 2021. Regional results indicate a significant positive trend in snow accumulation on glaciers in the Alps and Western Canada for the MERRA-2 and ERA-5 season-independent GBR respectively, non-significant trends in Scandinavia, and a significant positive trend of precipitation from the original MERRA-2 reanalysis (Tab. 4). In terms of air temperature, we observe a highly significant increase in the Alps

and Scandinavia. For single glaciers, the GBR models often indicate significant positive SWE trends in the Alps, significant





negative trends in Scandinavia, mostly non-significant trends in Central Asia, and significant negative to significant positive trends in Western Canada, depending on the elevation of the considered elevation range on the glaciers (Fig. 11b). However, the reliability of the GBR models to reproduce the relative changes of $B_w$ among years could not be robustly evaluated for glaciers with a low number of years with available $B_w$ data (see Fig. 7a). A high accuracy of the trends is thus not guaranteed

for these glaciers. This is especially important for many glaciers in Central Asia, where we often found low correlations, as well as many glaciers in Canada, where high but non-significant correlations were inferred in many cases (Fig. 11a).

Considering all investigated glaciers, the computed trends are generally positively correlated with elevation (Fig. 12a), i.e. glaciers at lower elevations show negative trends while glaciers at higher elevations often exhibit positive trends in SWE. However, for single regions (e.g. Central Asia) we find an overall negative correlation between the ERA-5 downscaled air

temperature trends and the GBR trends (Fig. 12b). This is also clearly noted for Western Canada, where the regional climate of glaciers located in the North-West is characterized by positive air temperature trends and negative SWE trends, while glaciers in the South-West and South (also located at higher average elevations) show less positive or even negative air temperature trends and positive SWE trends. These results suggest that SWE at high altitudes is changing differently depending on the region and on the elevation. Furthermore, despite SWE trends might be insignificant at the regional scale, they can be significant for

individual glaciers, thus emphasizing the importance of local-scale studies to evaluate SWE trends in high-mountain regions.

## 5   Discussion

The GBR models developed, evaluated and presented in this study showed a better overall agreement in terms of bias, spatial and temporal correlation with the $B_w$ data than the original reanalyses and the benchmark (lapse-rate-based approach described in Sec. 3.2.1) for the majority of the studied glaciers in the Alps, Scandinavia, Central Asia and Western Canada. SWE trends

(1981 to 2021) derived by applying our newly developed GBR models are more pronounced than those obtained from the total precipitation of the original reanalyses. On a regional scale, significant 41-year SWE trends are observed in the Alps (MERRA-2 season-independent GBR: +0.4 %/year) and in Western Canada (ERA-5 season-independent GBR: +0.2 %/year), while significant positive/negative trends are observed in all the regions for single glaciers. In the following, a comprehensive discussion of the approach and the results is provided.

### 5.1   Advantages and disadvantages of gradient boosting regressors

With the exception of some specific sites, our GBR models outperformed the benchmark method (lapse-rate-based approach (Sec. 3.2.1)) in the Alps, Scandinavia, Central Asia and Western Canada regarding the reduction of the bias against glacier-wide $B_w$ data (Figs. 7 and 8). This suggests that complex models such as our GBRs are needed to adjust reanalysis to different glaciers sites, which can be characterized by different topographical and climatic conditions, and where the performance of

reanalysis' estimates can vary greatly depending on the region (e.g. Sun et al., 2018). In fact, (independent) $B_w$ data was used to train our GBR models, allowing the GBRs to learn specific characteristics of actual snow accumulation on glaciers, and to transfer them to unknown sites (site-independent GBRs) and unknown seasons (season-independent GBRs).





The GBR models also outperform the benchmark to reproduce the spatial variability of the snow accumulation on single glaciers. $B_w$ data indicate decreasing SWE in the uppermost sections of many glaciers which may be attributed to preferential

snow deposition redistribution processes, caused by the interplay between snow, wind and the generally steep topography (e.g. Sold et al., 2016; Gerber et al., 2019). The use of downscaled air temperature and relative humidity as predictors (Fig. 4) and the ability of GBR models to model non-linear relationships allows a better representation of the vertical profiles of snow accumulation than the benchmark method (Fig. 10, Tab. 2). In fact, the observed spatial variability of $B_w$ could not be reproduced with the benchmark method, which by definition cannot represent decreasing values with the elevation (cf. Eq. 1).

Both, the GBR models and the benchmark do not require direct ground observations to be applied. However, the performance of the GBR models is influenced by the amount of data used to train the models. The impact of the data used to train the GBR models is discussed in the following section.

### 5.1.1   Spatial and temporal transferability of the GBR models

The GBR models were trained with almost 100 glaciers distributed over the four regions on three continents. Between the

regions we observed different robustness and performances. The performance of the GBR models tends to decrease when removing $B_w$ data of neighbouring glaciers from the training process (Fig. 9b, d, f and h). Neighbouring glaciers were removed from the training as a function of the distance (range) from the validated glacier in the cross-validation. Our results suggest that more available glaciers with $B_w$ data would probably greatly improve the performance in Central Asia and Western Canada, where our dataset is limited in terms of number of monitored glaciers and the horizontal spacing between different sites is

considerable. In the Alps, the network of monitored glaciers is much denser. Thus, more glaciers are excluded from the training for shorter distances than in other regions, impacting the performance of the site-independent GBR models. When the range of excluded neighbouring glaciers is extended to 1000 km, a strongly reduced number of glaciers of the same region is still used in the training, meaning that the models are almost exclusively trained with the glaciers of the other regions (the site-independent GBR becomes almost a region-independent GBR). The climate conditions and the complexity of the weather processes can

be very different among the four investigated regions (and even within the individual regions). A region-independent model is thus not expected to provide accurate results. In Scandinavia, a linear precipitation gradient with elevation is more appropriate than in the more complex topography of the Alps and Central Asia (e.g. Rasmussen and Andreassen, 2005). Thus, the site-independent GBR models are only performing slightly better than the benchmark when the full set of the other glaciers is used in the training, indicating that a simpler lapse-rate-based approach might be preferable. However, considering the four regions,

the bias of the region-independent GBR models remains comparable to the bias obtained with the benchmark method, which is independent from any ground observation.

The performance of the season-independent GBR models improved consistently when including in the training only a few other seasons of $B_w$ data related to the validated glacier in the cross-validation (Fig. 9a, c, e and g). This thus demonstrates the uniqueness of the snow accumulation distribution over each glacier that cannot be easily reproduced by using the relations

learned at other glaciers. However, this indicates that the snow accumulation distribution, and its relation with precipitation, is





similar in different years (see also e.g. Grünewald et al., 2013; Sold et al., 2016). For our application, there would be added benefit from $B_w$ data on additional glaciers rather than on additional seasons.

### 5.1.2 Impact of the chosen reanalyses on the GBR models

At a regional scale, the total precipitation estimated in the accumulation season by the original MERRA-2 has shown larger

biases than the original ERA-5 when compared to $B_w$ on glaciers. The coarser spatial resolution of MERRA-2 is certainly a factor causing larger biases in complex high-mountain areas (e.g. Zandler et al., 2019; Chen et al., 2021). In fact, a coarse resolution directly implies that mountains are more strongly smoothed. The absolute elevation of a grid cell is thus lower for a coarse resolution and the estimated precipitation also refers to the lower elevation of the grid cell.

The performance of the original ERA-5 and MERRA-2 has a direct impact on the GBR models. However, the GBR models

were able to compensate for such differences in the bias. In fact, the biases of the ERA-5 and MERRA-2 GBR models are much closer to each other than the biases of the original reanalyses (see Fig. 6a, b, c and d). The differences between the performance of our GBR models are also caused by the different predictors that have been used. For instance, the ERA-5 GBR models use more predictors describing the complexity of the grid cell topography than the MERRA-2 GBR models (see Fig. 4, Tab. B1 and B2).

### 450  5.1.3 Influence of the winter mass balance data accuracy on the GBR models

Our study strongly relies on observed snow accumulation data on glaciers. However, various problems are related to the direct measurements of snow accumulation on glaciers thus leading to uncertainties in the observations (e.g. Zemp et al., 2013; Sold et al., 2016; O'Neel et al., 2019; Huss et al., 2021). Most importantly, snow accumulation measured at individual points needs to be extrapolated in space to obtain $B_w$ data used in our analysis. At the highest elevations of glaciers with a typically

difficult accessibility for manual observations, results are often purely based on extrapolation techniques (e.g. Østrem and Brugman, 1966; Cogley et al., 2011; Huss et al., 2021). Given that the WGMS database does not generally report how this was achieved and how many actual observations were available in a given elevation interval, it is difficult to assess the integrative uncertainty in the $B_w$ data used. In order to illustrate the importance of the extrapolated $B_w$ data used in this study, we more closely inspected point winter snow observations for 12 Swiss sites and three years (2016-2018) based on a dataset with higher

resolution and full documentation (GLAMOS, 2021).

Figure 13a indicates that a lower number of manual observations was typically performed at the lowest and the highest elevations of the glaciers. In some elevation bands, even no manual observations are available and $B_w$ data refer to an extrapolation. However, a much larger number of manual observations is typically performed in the elevation intervals corresponding to the largest areas of the glaciers (Fig. 13a and c). As indicated by Fig. 13e, considerable uncertainties might exist in the analyzed

vertical profiles of $B_w$. However, the weighting function dependent on the area of the intervals used in the training of the GBR models assigned more importance to the $B_w$ data in such observation-rich areas. Furthermore, the main results of the study relate to glacier-wide $B_w$ data, which for most glaciers is very close to the glacier-wide mean of the manual observations as





also indicated in Fig. 13e (only in three out of 34 cases the ratio of glacier-wide mass balance to the average of all individual observations is larger than 1.10 and in no case the ratio is lower than 0.90).

Another source of uncertainty that is difficult to assess is the starting date of the accumulation season. We considered the same starting date for all elevation intervals, even though it varies over the glacier's elevation range. The accumulation of snow starts later at low elevations and earlier at high elevations. Therefore, the different elevations also collect different precipitation totals (as the periods differ). The impact on the study of the date considered as beginning of the accumulation season has been evaluated with a sensitivity test (Fig. 13). For the same 12 Swiss glaciers and three years as above we rely on the more detailed

data set of point winter mass balance data that documents start dates of measured cumulative snow precipitation of the winter season for each location individually. Start dates have been inferred based on a distributed glaciological modelling approach driven by daily local weather data (Huss et al., 2021). The total precipitation of ERA-5 and MERRA-2 were derived over these varying starting dates and were compared with the total precipitation obtained with non-varying, average starting dates (Fig. 13b, d and f). Figure 13b indicates that at high (low) elevations, the accumulation season can start up to 20 days before (after)

the unique date that we considered for all elevation intervals. These differences may translate in different amounts of total precipitation. In extreme cases, the total MERRA-2 (Fig. 13d) or ERA-5 (Fig. 13f) precipitation that would be obtained with varying dates would be almost twice (or half) of the total precipitation that we considered. However, the impact on the main results presented above is limited because these large differences are typically observed at the highest/lowest elevations of the glaciers, where the glacier area is minor and thus, a lower weight is assigned to the $B_w$ data in the training of the GBR models.

Moreover, the main results of the study are based on glacier-wide values and for both MERRA-2 and ERA-5 we observe very small differences in terms of glacier-wide precipitation totals for the majority of the Swiss glaciers. Only in six out of 34 cases the glacier-wide ratio is smaller (larger) than 0.97 (1.03), and only in two cases it exceeds 1.10.

    This analysis suggests that the used $B_w$ data and the considered start dates can lead to relevant uncertainties in the analysis of vertical profiles. However, this does not generally have a relevant impact on our conclusions, which are mainly based on

glacier-wide values.

## 5.2   SWE trend analysis and relation to precipitation and temperature changes

Rising air temperatures and related shifts from solid to liquid precipitation lead to an upward migration of the snowlines (e.g. Beniston et al., 2018) and shorter accumulation seasons (e.g. Marty, 2008), rather due to earlier snowmelt than due to later onset of the snowy season (Klein et al., 2016). In our study, we firstly (see Sec. 3.2.2) adjusted the estimates of the reanalyses

with respect to $B_w$ data. Since liquid precipitation is expected to refreeze in the snowpack (e.g. Wright et al., 2007), the potentially increasing ratio of liquid to solid precipitation will not immediately affect the SWE (except for a later beginning of the accumulation season). Secondly (see Sec. 3.4), we defined a fixed period and adjusted the total precipitation for that period, reducing the potential influence of later accumulation and earlier snowmelt. Thus, in the trend analysis, the GBR estimates represent the SWE as if the accumulation season would start on the first of October and continue until the 30th of April. The

potential influence of a later onset is indirectly accounted by the GBR models, which will result in a smaller adjustment factor when the total precipitation is compared with a smaller $B_w$.





The analysis of SWE trends is challenging due to their mutual dependency with temperature and precipitation changes. Generally, increasing precipitation will cause increasing SWE, while increasing temperature above the freezing point will cause decreasing SWE, therefore, the balance between temperature and precipitation changes will determine whether SWE trends will increase or decrease (Stranden and Skaugen, 2009). Hereafter, we discuss SWE trends on the 30th of April at high elevations obtained with our GBR models in relation to previous studies in the Alps, Scandinavia, Central Asia and Western Canada.

As shown by Fig. 12b, in the Alps, we often observed positive trends of both temperature and glacier-wide SWE (GBR estimates), suggesting that at high elevations, increased winter precipitation dominates over increased temperature. Most previous studies show rather negative trends in SWE, which are typically more pronounced at low elevations (e.g. Durand et al., 2009; Terzago et al., 2013). Decreasing trends are also observed in spring SWE (e.g. Bocchiola and Diolaiuti, 2010; Marty et al., 2017). However, in our study, we are focusing exclusively on the highest elevation of mountain ranges that are occupied by glaciers, where the SWE changes might be driven more by precipitation changes rather than increasing temperatures. Long-term time series (since 1914) of seasonal mass-balance observations on three Swiss glaciers (Claridenfirn, Grosser Aletschgletscher and Silvrettagletscher) were analyzed by Huss and Bauder (2009), who observed winter balances below the long-term average in the period 1982-2007, partly related to a prolongation of the melting season. They also evaluated the annual fraction of solid precipitation compared to total precipitation, observing a significant decrease of the solid precipitation fraction after 1970. The differences to our study can be explained by the longer time interval extending beyond 1981 as in the present study.

A decrease in spring SWE is typically observed at low elevations in Norway (e.g. Skaugen et al., 2012). Generally, regions characterized by colder winter climate, show positive long-term trends, with variation in snow depth mainly linked to variations in precipitation. In regions of warmer winter climate, the variation in snow depth is dominated by temperature, and long-term trends are mainly negative (e.g. Dyrrdal et al., 2013). In our study, we do not observe a clear relation between elevation or air temperature changes and SWE trends of different glaciers (Fig. 12a and b). We only observe significantly negative SWE trends. However, we observe stronger decrease of SWE at low elevations than at high elevations for single glaciers (e.g. Nigardsbreen in Fig. 11b). Furthermore, despite regional precipitation trends were found to be positive over the past century (e.g. Hanssen-Bauer, 2005), the seNorge model in Norway (see Supplementary Fig. S1) indicates that precipitation trends are slightly decreasing since 1981 over the majority of glaciers contained in our study, which further supports the negative SWE trends inferred.

For Central Asia Smith and Bookhagen (2018) examined SWE trends using passive microwave data (1987 to 2009) and found an overall decrease in snow accumulation in High Mountain Asia, although regions with increased SWE in the Pamir, Kunlun Shan, Eastern Himalaya, and Eastern Tien Shan were detected. Our study does not provide a uniform picture for Central Asia. Mostly insignificant trends are found and the ability of the GBR models to reproduce the temporal variability of the snow accumulation over years could not be proven for many glaciers (e.g. Kara-Batkak glacier (Fig. 11b)), thus making the derived trend somewhat unreliable.





For a study in the Central Rocky Mountains, Sospedra-Alfonso et al. (2015) estimated a threshold elevation (1560 ± 120 m), below (above) which temperature (precipitation) is the main driver of SWE and snowpack duration. The trends obtained with the GBR models over the Canadian glaciers are mainly positive for ERA-5, while the MERRA-2 GBRs indicated both negative and positive trends that seem to be clustered as function of the glacier's mean elevation (Fig. 12b). Our results are thus in line with previous studies in North America (e.g. Mote, 2003, 2006; Mote et al., 2018; Howat and Tulaczyk, 2005), and also confirm that increasing temperatures at the elevation of glaciers usually imply negative or slightly positive SWE trends (Fig. 12b).

The trends obtained in the four investigated regions confirm that at the highest elevations, the SWE tends to increase more than at lower elevations on glaciers, where it is often decreasing, because of the effect of increasing temperatures. Our results show highly significant SWE trends when focusing on individual glaciers due to local climatic differences. However, significant regional trends are only observed for the MERRA-2 GBR model in the Alps (positive) and Scandinavia (negative) (see Table 4). Thus, the results emphasize the importance of performing a local analysis for changes of snow and precipitation in high-mountain regions.

## 6 Conclusions

In this study, we developed and evaluated a machine learning approach based on gradient boosting regressor models to adjust the total precipitation of reanalysis datasets (ERA-5 and MERRA-2) over the accumulation season on glaciers. The high performance achieved with our approach allowed us to use it to derive observation-independent SWE estimates over glaciers in the Alps, Scandinavia, Central Asia and Western Canada. Data on snow accumulation distribution at the end of the accumulation season covering a period of up to 41 years from 95 glaciers (Zemp et al., 2021) were used to train our approach.

The most important variables that were automatically selected by our GBR models were those related to the elevation difference between the glacier surface and the terrain model underlying the reanalyses, as well as the downscaled air temperature and relative humidity. The latitude and longitude of the studied sites were also frequently used in order to discriminate between regions that are characterized by different climate conditions and weather systems, allowing the GBR models to be split into individual sub-models adapted to specific sub-regions.

In general, the total precipitation of the reanalyses largely underestimates observed snow accumulation on glaciers. The largest (relative) regional underestimation is observed in Central Asia for MERRA-2 and in Scandinavia for ERA-5 (Fig. 6). The GBR models allowed reducing these biases. In Central Asia and Western Canada, the correlation between the original reanalyses' estimates and the snow accumulation on the analyzed glaciers has considerably increased with the season-independent GBRs only. With the exception of some specific glaciers, our GBR models outperformed the benchmark method (lapse-rate-based approach) in the Alps, Scandinavia, Central Asia and Western Canada by reducing the bias of the original re-analysis against the $B_w$ data (Fig. 7). This suggests that complex models such as our GBRs are needed to adjust reanalysis data to different glaciers that can be characterized by different topographical and climatic conditions, and where the performance of reanalysis' estimates can vary greatly depending on the region.





The results also indicate that the season-independent GBRs outperform the site-independent GBRs to reduce the bias, con-

firming that filling data gaps is much simpler than estimating SWE on glaciers with no observations. Thus, denser ground-based

or improved remote sensing observations, with the further development of methods that guarantee the spatio-temporal trans-

ferability of the observed snow and/or precipitation in high-mountain areas, would be very important.

The GBR models have also shown improved performance than the original reanalyses to reproduce temporal changes (over

years) of snow accumulation on the majority of the analyzed glacier sites. We thus derived SWE trends over glaciers, obtaining

more enhanced trends from the GBRs than from the original reanalyses. According to our season-independent GBR models, the

SWE is significantly increasing over the majority of glaciers in the Alps (region-wide trend: +0.4 %/year for MERRA-2), and

in Western Canada (region-wide trend: +0.2 %/year for ERA-5). When considering individual glaciers, mostly non-significant

trends are found for glaciers in Central Asia, often highly significant negative trends in Scandinavia and latitude/elevation-

dependent trends in Western Canada, thus emphasizing the importance of local studies when analyzing snow and precipitation

in high-mountain regions.

We demonstrated that machine learning models (with robust cross-validation schemes) can be powerful instruments to adjust

precipitation estimates over glaciers. The new information provided by our study can be helpful to further evaluate the local

impact of climate change over glaciers in different regions of the world, where observations are often scarce and the spatial

resolution of the reanalysis products is too coarse to allow local impact studies and the consequent development of adaptation

strategies.

**Appendix A: Equations used to derive the relative humidity**

The relative humidity is not directly provided by all the reanalysis products, therefore we derived it by applying a similar

approach to Liston and Elder (2006) and Gupta and Tarboton (2016), which is presented hereafter.

**A1    ERA-5**

The relative humidity is not directly provided at the grid level, therefore, we combined the 2 m temperature ($t2m$) and dew

point temperature ($d2m$) as follows:

$$r2m^* = \frac{a * exp(\frac{b*d2m}{c+d2m})}{a * exp(\frac{b*t2m}{c+t2m})} \tag{A1}$$

where $r2m^*$ is the computed 2 m relative humidity and for ice/snow, $a = 611.21\,Pa$, $b = 22.452$ and $c = 272.55\,°C$.

**A2    MERRA-2**

MERRA-2 is not providing the relative humidity at the grid and at the pressure levels, furthermore, the dew point temperature

is not provided at the pressure levels either, therefore, we combined the specific humidity and the pressure in order to derive

them (at the grid and at the pressure levels). For ice/snow, $a = 611.21\,Pa$, $b = 22.452$ and $c = 272.55\,°C$.



Vapour pressure:

$$e^* = \frac{QV * P}{0.622 + QV} \tag{A2}$$

where the specific humidity $QV = QV10M$ for the grid and $QV = QV_{\text{levels}}$ for the pressure levels, the pressure $P = PS$ for the grid and $P = P_{\text{levels}}$ for the pressure levels. The vapour pressure $e^*$ was named $e10M^*$ for the grid and $e^*_{\text{levels}}$ for the pressure levels.

Dew point temperature:

$$Td^* = 273.15 + \frac{c * \ln\left(\frac{e^*}{a}\right)}{b - \ln\left(\frac{e^*}{a}\right)} \tag{A3}$$

The dew point temperature $Td^*$ was named $Td10M^*$ for the grid and $Td^*_{\text{levels}}$ for the pressure levels.

Relative humidity:

$$RH^* = \frac{a * exp\left(\frac{b*Td^*}{c+Td^*}\right)}{a * exp\left(\frac{b*T}{c+T}\right)} \tag{A4}$$

where the temperature $T = T10M$ for the grid and $T = T_{\text{levels}}$ for the pressure levels. The relative humidity $RH^*$ was named $RH10M^*$ for the grid and $RH^*_{\text{levels}}$ for the pressure levels.

**Appendix B: Derivation and list of the variables used in the models**

In Table B1 and Table B2, we report the complete list of variables selected from the reanalyses products. In Table B3 we provide a summary of all the variables used by the GBR models.

*Author contributions.* MGu conducted the analysis and wrote the manuscript with inputs from all co-authors. MH derived the snow accumulation data for the Swiss glaciers analyzed in Section 2.2 and 5.1.3 and contributed to the related discussion. MGa and NS contributed to the design of the research and helped with continuous discussions concerning the results. All co-authors contributed to the final form of the manuscript.

*Competing interests.* The authors declare that they have no competing interests.

*Acknowledgements.* The study is part of the High-SPA 200021_178963 project, which is funded by the Swiss National Science Foundation (SNSF). We would like to acknowledge the WGMS and all the groups providing freely available in-situ observations on glaciers. The authors would also like to acknowledge NASA for providing the freely available MERRA-2 products (Global Modeling and Assimilation Office (GMAO), 2015a, b, c). Hersbach et al. (2018a, b) were downloaded from the Copernicus Climate Change Service (C3S) Climate Data



Store. The results contain modified Copernicus Climate Change Service information 2020. Neither the European Commission nor ECMWF

is responsible for any use that may be made of the Copernicus information or data it contains.



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





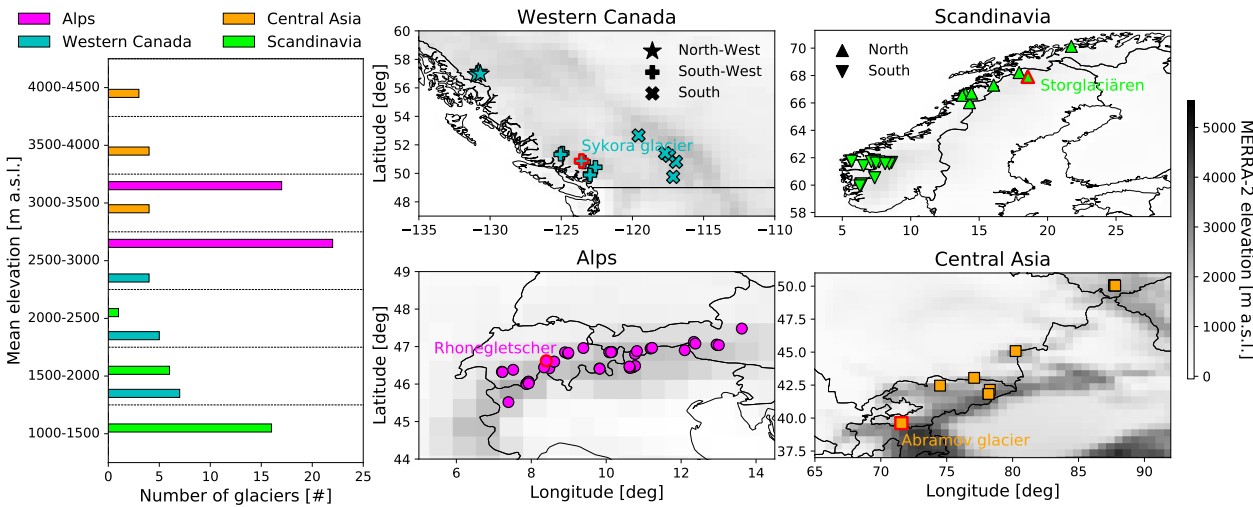

**Figure 1.** Mean elevation and distribution of the glaciers used in the study (data source: (Zemp et al., 2021)). Glaciers shown in Fig. 10 are highlighted in red.



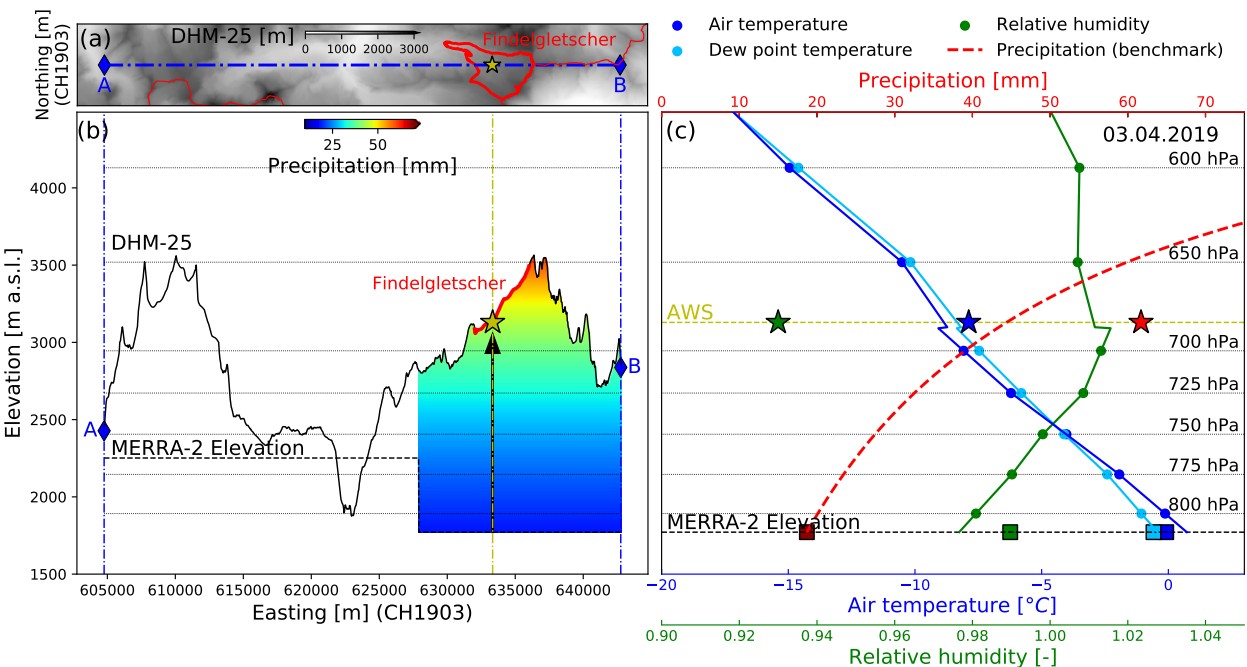

**Figure 2.** The downscaling methods for daily precipitation using the benchmark method (cf. Sec. 3.2.1)), air temperature and relative humidity at the example of Findelgletscher, Southern Switzerland. (a) High-resolution topography in the region of Findelgletscher. (b) The elevation of the MERRA-2 grid is compared with the elevation of a detailed terrain model, showing a longitudinal cross-section of the glacier (bold red line) and the surrounding regions. (c) Resulting downscaled daily variables on 3 April, 2019. The stars indicate the observations by an automatic weather station (AWS) located on the glacier. The measured precipitation refers to the daily difference of SWE measured by a cosmic ray sensor (see Gugerli et al., 2019).

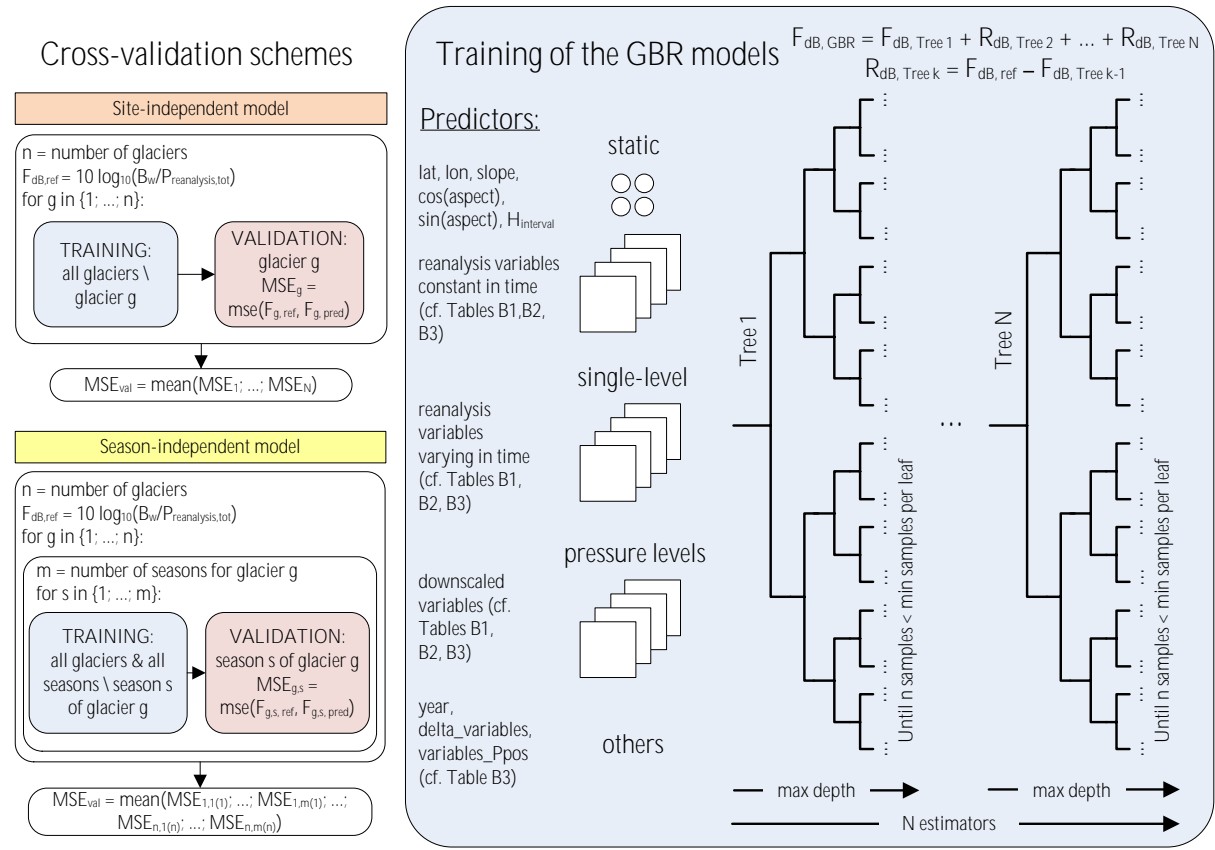

**Figure 3.** Cross-validation schemes for the site-independent and season-independent GBR models, and schematic representation of the training process and the predictors used by the models, where $F_{\mathrm{dB,ref}}$ is the reference adjustment factor (Eq. 2), $R_{\mathrm{dB},\,Tree\,k}$ is the difference between $F_{\mathrm{dB,ref}}$ and the adjustment factor estimated by the "$k-1$" tree during the training ($F_{\mathrm{dB},\,Tree\,k-1}$) and $F_{\mathrm{dB,GBR}}$ is the adjustment factor estimated by the full GBR model.



**Figure 4.** Frequency of use of the ten most used predictors by the site-independent GBR models based on (a) ERA-5, and (b) MERRA-2. (c) and (d) show the principal component analysis (PCA) biplots for the site-independent GBR models based on ERA-5 and MERRA-2, considering the ten most used predictors of (a) and (b). The PCA projects the set of ten predictors into a new space characterized by the principal components, which are uncorrelated and orthogonal. Each principal component is the result of a linear combinations between the predictors and the first component explains the largest part of the predictors' variance. The average adjustment factor between $B_w$ and reanalysis' precipitation of each glacier depending on the two first principal components is presented. The meaning of the predictors' names is described in Tables B1 (original ERA-5 variables), B2 (original MERRA-2 variables) and B3 (downscaled ERA-5 and MERRA-2 variables). A predictor characterized by a longer arrow has a higher weight in the principal component. The correlation between the predictors can be inferred by looking at the angle between the arrows. The position of the orthogonal projection of the sites on the arrows is proportional to the value of the predictor for the site with respect to the other sites.

**Figure 5.** Principal component analysis (PCA) biplots for the regional site-independent GBR models, considering the ten most used predictors of (a, c, e, g) ERA-5 and (b. d. f. h) MERRA-2, similarly to Fig. 4. The colors represent the factors between B$_w$ and the total precipitation of ERA-5 and MERRA-2, respectively. The meaning of the predictors' names is described in Tables B1 (original ERA-5 variables), B2 (original MERRA-2 variables) and B3 (downscaled ERA-5 and MERRA-2 variables).



**Figure 6.** Comparison between all glacier-wide $B_w$ values and the model estimates: (a) original MERRA-2 , (b) original ERA-5, (c) MERRA-2 site-independent GBR, (d) ERA-5 site-independent GBR, (e) MERRA-2 season-independent GBR, (f) ERA-5 season-independent GBR. The Pearson correlation (CORR), the root-mean-square error (RMSE), the fraction of standard error (FSE, corresponding to the RMSE divided by the regional mean $B_w$) and the number of all seasons of all glaciers (N pts) for each region are also reported. The boxplots indicate the distribution of the model's estimates (right) and of the $B_w$ data (top) for each region.





**Figure 7.** (a) Number of seasons with available $B_w$ data for each glacier. Factors between seasonal glacier-wide $B_w$ and (b) ERA-5-based models and (c) MERRA-2 based models, for each glacier of the study. The variability shown in the boxplots is given by the different seasons of $B_w$ data.



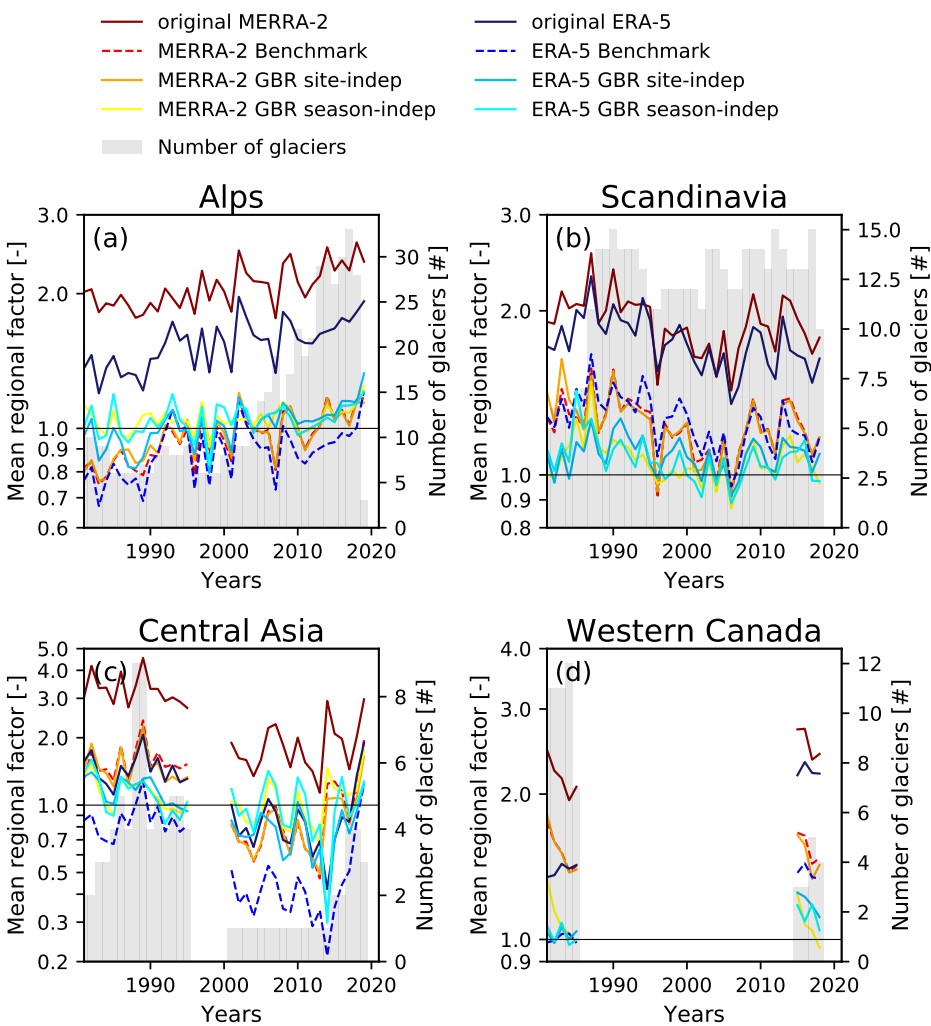

**Figure 8.** Mean regional factor between the $B_w$ data and the reanalysis-based models' estimates as a function of the accumulation seasons from 1981 to 2019 (last available season): (a) Alps, (b) Scandinavia, (c) Central Asia and (d) Western Canada. For each season, all the glaciers with available $B_w$ data were considered (the number of glaciers used to derive the regional factor is indicated by the gray bars).



**Figure 9.** Evaluation of the mean regional factor between the $B_w$ data and the reanalysis-based models' estimates ($F_{dB, region}$ defined in Eq. 6) depending on different data used in the training of the GBR models. Left column - evaluation of the performance of the season-independent GBR models as a function of the number of seasons used for training (of the validated glacier in the cross-validation). Training based on data of a varying number of available accumulation seasons. (a) Model validation depending on the number of training seasons per glacier in the Alps, (c) Scandinavia, (e) Central Asia and (g) Western Canada. Right column - evaluation of the robustness of the GBR models as a function of the number of other glaciers in the same region used in the training. All glaciers located within a range growing from 0 to 1000 km (from the validated glacier in the cross-validation) were excluded from the training. (b) Model validation depending on the range of excluded glaciers from the training in the Alps, (d) Scandinavia, (f) Central Asia and (h) Western Canada.



**Figure 10.** Vertical profiles of $B_w$ data and the modelled SWE at the end of a specific accumulation season: (a) Rhonegletscher (Alps), (b) Abramov glacier (Central Asia), (c) Storglaciären (Scandinavia) and (d) Sykora glacier (Western Canada).

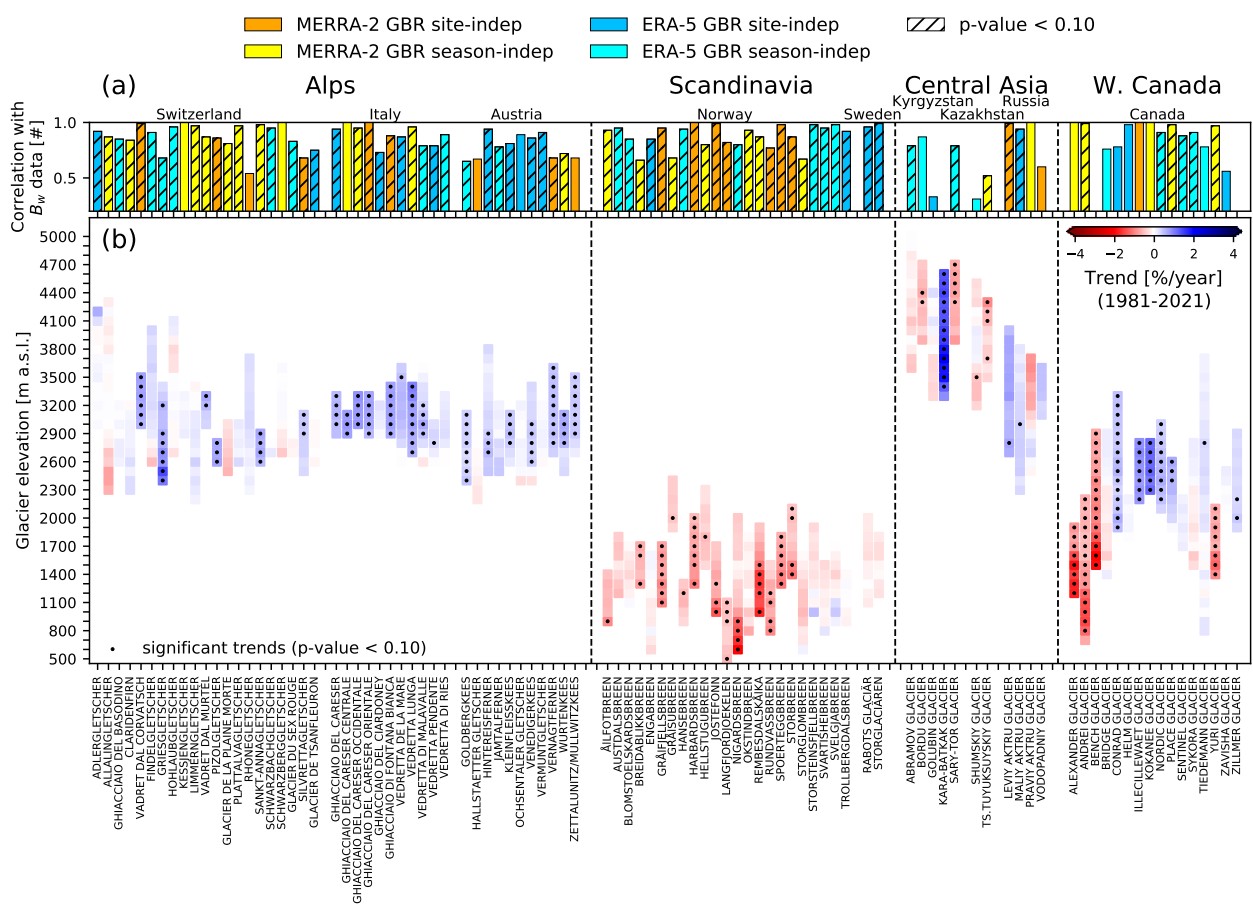

**Figure 11.** 41-year SWE trends obtained with the GBR models on all glaciers analyzed in the study. (a) Pearson correlation between GBR models and glacier-wide $B_w$ over the accumulation seasons (temporal correlation). For each glacier, only the correlation of the GBR with the highest correlation is shown. The significance of the correlation is based on Student's t-distribution. (b) Relative SWE trends on the 30th of April for the period 1981-2021 obtained with the GBR model with the highest correlation with $B_w$ data. The trends are relative to the mean of the respective GBR estimate for the considered glacier over the entire period (1981-2021).

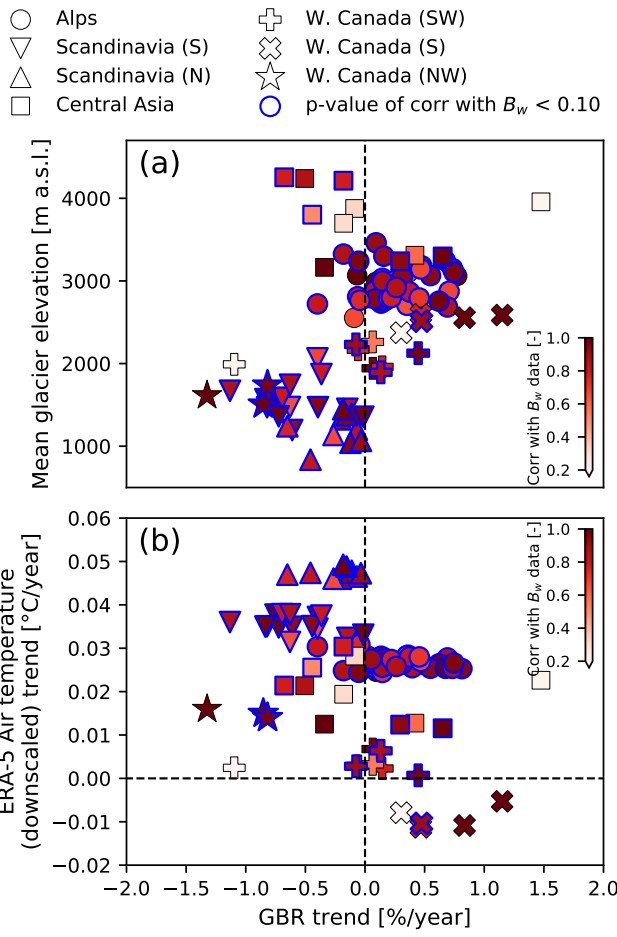

**Figure 12.** Relation between the GBR-derived SWE trends and elevation/temperature changes on all glaciers analyzed in the study. (a) Comparison between the mean elevation of the glaciers and the relative SWE trends obtained with the GBR models. (b) Comparison between the absolute trends in downscaled air temperature of ERA-5 and the relative SWE trends obtained with the GBR models. The GBR models with the highest correlation with the glacier-wide $B_w$ over the accumulation seasons (temporal correlation) are used. The significance of the correlation is based on Student's t-distribution.



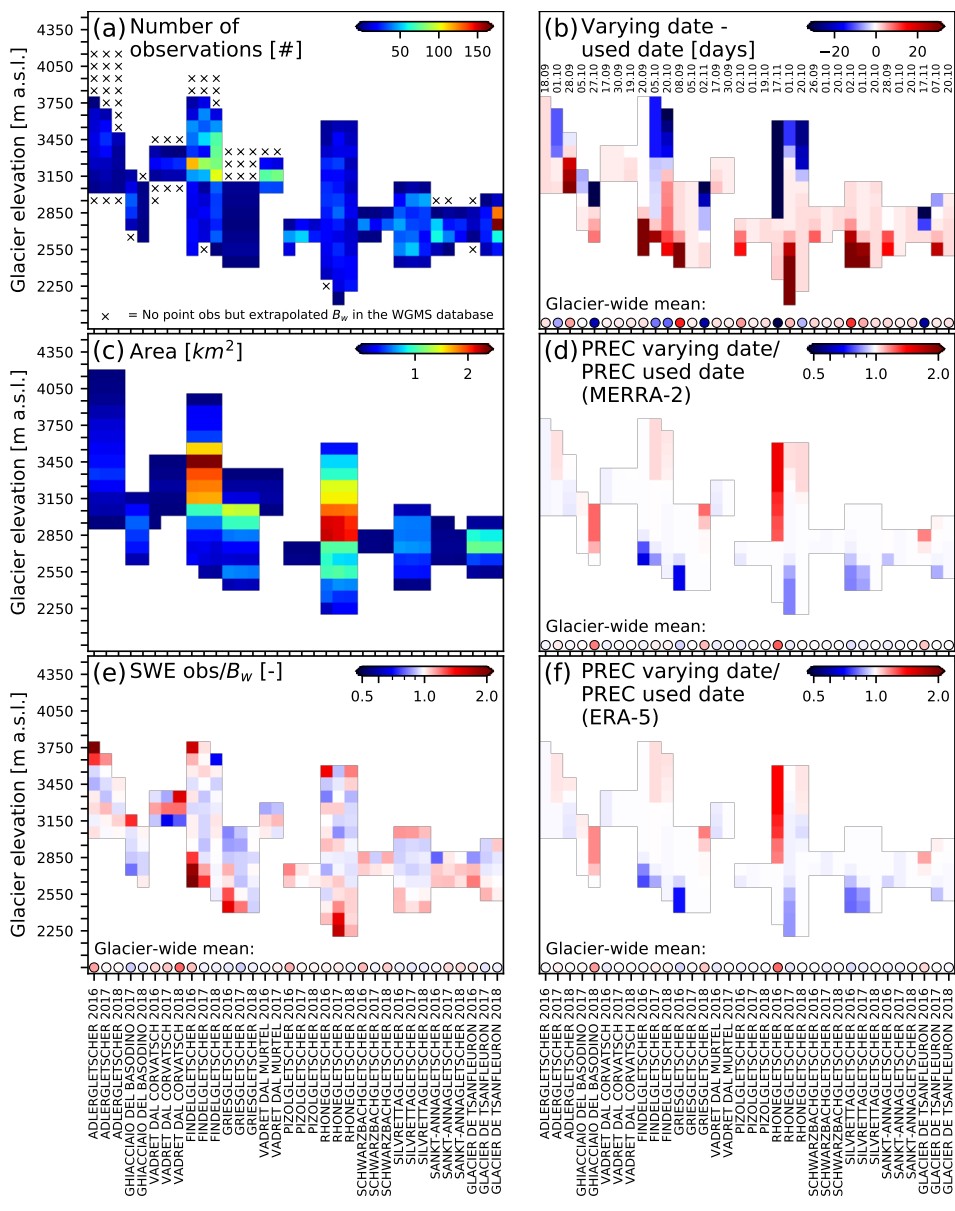

**Figure 13.** Sensitivity analysis of extrapolated $B_w$ data and used starting dates for 12 glaciers in the Swiss Alps between 2016 and 2018 (GLAMOS, 2021). Left column - difference between $B_w$ data used in this study and point-SWE observations: (a) Number of manual observations performed in the elevation intervals of the glaciers. (c) Area of the glacier according to the elevation interval. (e) Ratio between the observed SWE and the $B_w$ data. Right column - impact of the date considered as beginning of the accumulation season on seasonal precipitation totals: (b) Differences between accurate (varying) dates of the beginning of the accumulation period and the used dates in the study (the day and the month of the used dates are written on the figure (DD.MM)). (d) Ratio between the total precipitation of MERRA-2 according to the accurate dates and the used dates. (f) Ratio between the total precipitation of ERA-5 according to the accurate dates and the used dates.





**Table 1.** Hyperparameters of the optimized GBR models. *Max. predictors* indicates the number of considered predictors when looking for the best split. *Subsample* indicates the fraction of samples used for fitting the individual learners.

|  | Site-independent GBR | | Season-independent GBR | |
|  | ERA-5 | MERRA-2 | ERA-5 | MERRA-2 |
| --- | --- | --- | --- | --- |
| n estimators | 100 | 100 | 90 | 90 |
| max. depth | 7 | 6 | 7 | 7 |
| min. samples leaf | 20 | 70 | 6 | 6 |
| learning rate | 0.05 | 0.05 | 0.10 | 0.10 |
| max. predictors | sqrt | sqrt | sqrt | sqrt |
| subsample | 0.8 | 0.8 | 1.0 | 1.0 |



**Table 2.** Scores of the reanalysis-based models against the $B_w$ along the vertical profiles. Here, we report the mean score of all the accumulation seasons. Only glaciers with a minimum of five seasons and a minimum of ten elevation intervals of $B_w$ data per season are reported. The Pearson correlation (r [-]) and the coefficient of variation (CV [%], corresponding to the percentage ratio of the standard deviation to the mean) are reported.

| | r (CV) | | | | | | CV | n seasons | avg n obs |
| | Benchmark | | GB site-indep | | GB season-indep | | $B_w$ | | |
| Glacier | ERA-5 | MERRA-2 | ERA-5 | MERRA-2 | ERA-5 | MERRA-2 | | | |
|---|---|---|---|---|---|---|---|---|---|
| Adlergletscher | -0.56 (28) | -0.60 (37) | 0.74 (14) | 0.70 (14) | 0.72 (25) | 0.72 (25) | 26 | 10 | 12 |
| Allalingletscher | -0.40 (43) | -0.46 (51) | 0.91 (23) | 0.90 (21) | 0.89 (22) | 0.91 (23) | 19 | 16 | 19 |
| Findelgletscher | 0.46 (30) | 0.40 (38) | 0.46 (12) | 0.36 (14) | 0.82 (22) | 0.83 (21) | 33 | 11 | 14 |
| Hohlaubgletscher | -0.74 (30) | -0.77 (35) | 0.89 (11) | 0.84 (11) | 0.92 (14) | 0.90 (14) | 23 | 8 | 13 |
| Rhonegletscher | 0.81 (27) | 0.79 (30) | 0.85 (23) | 0.86 (19) | 0.91 (30) | 0.88 (26) | 37 | 8 | 14 |
| Vedretta de La Mare | 0.76 (19) | 0.76 (19) | 0.26 (11) | 0.76 (8) | 0.82 (18) | 0.78 (19) | 25 | 14 | 18 |
| Vedretta Lunga | -0.24 (13) | -0.25 (15) | -0.06 (12) | -0.06 (11) | 0.06 (8) | 0.03 (9) | 14 | 13 | 14 |
| Vedretta Malavalle | 0.73 (17) | 0.73 (18) | 0.86 (9) | 0.87 (10) | 0.90 (21) | 0.89 (23) | 27 | 14 | 17 |
| Goldbergkees | -0.39 (15) | -0.39 (15) | -0.29 (10) | -0.29 (17) | -0.10 (5) | -0.22 (9) | 14 | 15 | 15 |
| Hallstaetter Gletscher | 0.95 (15) | 0.95 (15) | 0.73 (3) | 0.74 (2) | 0.96 (18) | 0.95 (21) | 32 | 6 | 14 |
| Hinterisferner | 0.31 (25) | 0.29 (28) | 0.82 (15) | 0.79 (13) | 0.83 (19) | 0.82 (20) | 26 | 7 | 25 |
| Venedigerkees | 0.90 (20) | 0.89 (21) | 0.71 (10) | 0.80 (11) | 0.90 (25) | 0.86 (21) | 32 | 6 | 20 |
| Vernagtferner | 0.52 (18) | 0.52 (19) | 0.18 (4) | 0.53 (2) | 0.67 (7) | 0.63 (6) | 15 | 31 | 17 |
| Wurtenkees | -0.16 (13) | -0.16 (13) | -0.19 (5) | -0.14 (5) | 0.05 (3) | 0.00 (5) | 18 | 26 | 13 |
| Zettalunitz/Mullwitzkees | -0.02 (16) | -0.03 (17) | 0.50 (5) | 0.38 (5) | 0.8 (11) | 0.70 (11) | 27 | 6 | 16 |
| Blomstoelskardsbreen | 0.96 (11) | 0.96 (11) | 0.97 (14) | 0.96 (16) | 0.96 (17) | 0.96 (16) | 23 | 10 | 12 |
| Graafjellsbreen | 0.97 (11) | 0.97 (11) | 0.97 (27) | 0.96 (21) | 0.96 (26) | 0.95 (23) | 25 | 10 | 12 |
| Harbardsbreen | -0.09 (13) | -0.09 (13) | -0.03 (28) | 0.01 (23) | -0.03 (19) | 0.08 (18) | 10 | 5 | 14 |
| Hellstugubreen | 0.88 (13) | 0.88 (14) | 0.91 (21) | 0.74 (7) | 0.92 (30) | 0.90 (25) | 34 | 37 | 14 |
| Nigardsbreen | 0.96 (26) | 0.96 (27) | 0.96 (51) | 0.96 (50) | 0.97 (52) | 0.97 (54) | 51 | 38 | 14 |
| Rembesdalskåka | 0.93 (13) | 0.93 (13) | 0.92 (30) | 0.93 (30) | 0.95 (38) | 0.95 (39) | 42 | 38 | 14 |
| Rundvassbreen | 0.98 (12) | 0.98 (13) | 0.94 (30) | 0.96 (28) | 0.95 (42) | 0.95 (44) | 49 | 8 | 13 |
| Storbreen | 0.94 (12) | 0.94 (12) | 0.90 (24) | 0.89 (21) | 0.93 (24) | 0.93 (25) | 27 | 30 | 13 |
| Svartisheibreen | 0.60 (12) | 0.60 (13) | 0.78 (21) | 0.78 (27) | 0.79 (19) | 0.82 (24) | 19 | 7 | 13 |
| Svelgjabreen | 0.97 (15) | 0.97 (15) | 0.96 (35) | 0.98 (29) | 0.98 (32) | 0.98 (32) | 31 | 10 | 16 |
| Rabots Glaciaer | 0.95 (14) | 0.95 (14) | 0.95 (39) | 0.97 (31) | 0.96 (35) | 0.96 (36) | 37 | 6 | 30 |
| Storglaciären | 0.88 (10) | 0.88 (11) | 0.78 (25) | 0.71 (19) | 0.89 (28) | 0.87 (29) | 44 | 5 | 28 |
| Abramov glacier | 0.13 (28) | 0.10 (31) | 0.68 (12) | 0.81 (20) | 0.96 (35) | 0.96 (39) | 40 | 14 | 13 |
| Golubin glacier | 0.35 (22) | 0.34 (23) | 0.53 (10) | 0.78 (12) | 0.84 (46) | 0.85 (36) | 56 | 8 | 22 |
| Shumskiy glacier | -0.48 (25) | -0.49 (28) | -0.02 (17) | 0.26 (14) | 0.53 (12) | 0.58 (15) | 24 | 7 | 39 |
| Ts. Tuyuksuyskiy glacier | -0.49 (15) | -0.50 (16) | 0.78 (11) | 0.75 (9) | 0.81 (14) | 0.75 (15) | 19 | 31 | 12 |
| Leviy Aktru glacier | 0.29 (28) | 0.28 (28) | 0.72 (19) | 0.70 (13) | 0.80 (23) | 0.69 (24) | 26 | 7 | 13 |
| Maliy Aktru glacier | 0.80 (29) | 0.79 (29) | 0.77 (30) | 0.76 (25) | 0.82 (38) | 0.82 (38) | 48 | 7 | 15 |
| Bench glacier | 0.76 (28) | 0.76 (28) | 0.82 (25) | 0.84 (22) | 0.88 (35) | 0.88 (26) | 36 | 5 | 14 |
| Bridge glacier | 0.92 (30) | 0.92 (29) | 0.97 (41) | 0.97 (38) | 0.97 (43) | 0.97 (42) | 44 | 5 | 16 |
| Sykora glacier | 0.92 (26) | 0.92 (26) | 0.94 (39) | 0.94 (37) | 0.93 (39) | 0.95 (38) | 37 | 5 | 14 |
| Tiedemann glacier | 0.68 (60) | 0.70 (57) | 0.93 (41) | 0.89 (44) | 0.96 (40) | 0.92 (45) | 36 | 5 | 26 |

The original ERA-5 and MERRA-2 are not reported since they do not vary along the vertical profile of the glacier (a unique grid cell is assigned to the glacier).





**Table 3.** Pearson correlation (r [-]) between the reanalysis-based models and the glacier-wide $B_w$ over the accumulation seasons (temporal correlation). Only the glaciers with a minimum of 10 seasons with $B_w$ data are reported (i.e. no glacier in Western Canada). The significance of the correlation is based on Student's t-distribution.

| | r (p-value) | | | | | | | | n seasons |
|---|---|---|---|---|---|---|---|---|---|
| | Original | | Benchmark | | GB site-indep | | GB season-indep | | |
| Glacier | ERA-5 | MERRA-2 | ERA-5 | MERRA-2 | ERA-5 | MERRA-2 | ERA-5 | MERRA-2 | |
| Adlergletscher | **0.89** | 0.46 | **0.84** | 0.32 | **0.92** | **0.66** | **0.9** | **0.74** | 10 |
| Allalingletscher | **0.71** | **0.74** | **0.68** | **0.7** | **0.72** | **0.83** | **0.79** | **0.87** | 16 |
| Ghiacciaio del Basodino | **0.77** | **0.69** | **0.79** | **0.72** | **0.83** | **0.76** | **0.85** | 0.56 | 10 |
| Claridenfirn | **0.71** | **0.83** | **0.74** | **0.85** | **0.74** | **0.82** | **0.75** | **0.84** | 38 |
| Findelgletscher | **0.86** | 0.46 | **0.86** | 0.45 | **0.88** | 0.54 | **0.91** | **0.74** | 11 |
| Griesgletscher | **0.45** | **0.53** | **0.47** | **0.55** | **0.53** | **0.63** | **0.68** | **0.64** | 34 |
| Pizolgletscher | **0.77** | **0.84** | **0.77** | **0.84** | **0.78** | **0.86** | **0.7** | **0.79** | 10 |
| Silvrettagletscher | **0.56** | **0.63** | **0.55** | **0.62** | **0.54** | **0.68** | **0.61** | **0.65** | 35 |
| Ghiacciaio del Careser | **0.92** | **0.85** | **0.92** | **0.84** | **0.94** | **0.87** | **0.93** | **0.8** | 14 |
| Ghiacciaio del Ciardoney | **0.66** | **0.52** | **0.66** | **0.52** | **0.73** | **0.51** | **0.67** | 0.28 | 19 |
| Ghiacciaio di Fontana Bianca | **0.88** | **0.9** | **0.87** | **0.89** | **0.85** | **0.88** | **0.77** | **0.85** | 15 |
| Vedretta de La Mare | **0.79** | **0.76** | **0.71** | **0.68** | **0.87** | **0.82** | **0.8** | **0.85** | 14 |
| Vedretta Lunga | **0.94** | **0.9** | **0.94** | **0.91** | **0.95** | **0.95** | **0.94** | **0.96** | 13 |
| Vedretta di Malavalle | **0.57** | **0.64** | **0.62** | **0.7** | **0.76** | **0.77** | **0.79** | **0.76** | 14 |
| Vedretta Pendente | 0.37 | 0.47 | 0.39 | 0.49 | **0.79** | **0.74** | **0.66** | **0.69** | 14 |
| Goldbergkees | 0.46 | 0.31 | 0.46 | 0.31 | **0.62** | **0.57** | **0.65** | 0.45 | 15 |
| Jamtalferner | **0.6** | **0.65** | **0.6** | **0.64** | **0.67** | **0.72** | **0.78** | **0.69** | 24 |
| Kleinfleisskees | **0.7** | **0.63** | **0.72** | **0.64** | **0.81** | **0.67** | **0.71** | 0.46 | 14 |
| Vernagtferner | **0.64** | **0.66** | **0.64** | **0.65** | **0.62** | **0.68** | **0.48** | **0.51** | 31 |
| Wurtenkees | **0.65** | **0.56** | **0.65** | **0.56** | **0.72** | **0.63** | **0.71** | **0.72** | 26 |
| Aalfotbreen | **0.8** | **0.78** | **0.81** | **0.79** | **0.88** | **0.91** | **0.92** | **0.93** | 37 |
| Austdalsbreen | **0.94** | **0.91** | **0.94** | **0.91** | **0.95** | **0.95** | **0.95** | **0.92** | 31 |
| Blomstoelskardsbreen | **0.85** | **0.81** | **0.85** | **0.81** | **0.85** | **0.82** | **0.85** | **0.84** | 10 |
| Breidablikkbreen | 0.53 | 0.47 | 0.53 | 0.47 | 0.59 | 0.63 | 0.35 | **0.66** | 10 |
| Engabreen | **0.84** | **0.79** | **0.82** | **0.77** | **0.85** | **0.81** | **0.83** | **0.8** | 38 |
| Graafjellsbreen | **0.91** | **0.87** | **0.91** | **0.87** | **0.93** | **0.95** | **0.92** | **0.89** | 10 |
| Graasubreen | **0.38** | 0.28 | **0.38** | 0.28 | **0.52** | **0.54** | **0.54** | **0.68** | 38 |
| Hansebreen | **0.88** | **0.85** | **0.88** | **0.85** | **0.94** | **0.93** | **0.94** | **0.93** | 32 |
| Hellstugubreen | **0.59** | **0.46** | **0.58** | **0.44** | **0.67** | **0.67** | **0.58** | **0.8** | 37 |
| Langfjordjoekelen | **0.74** | **0.72** | **0.74** | **0.72** | **0.82** | **0.82** | **0.77** | **0.75** | 26 |
| Nigardsbreen | **0.76** | **0.72** | **0.75** | **0.71** | **0.8** | **0.79** | **0.8** | **0.76** | 38 |
| Okstindbreen | **0.93** | **0.94** | **0.93** | **0.94** | **0.93** | **0.88** | **0.91** | **0.93** | 10 |
| Rembesdalskåka | **0.78** | **0.74** | **0.74** | **0.7** | **0.81** | **0.84** | **0.84** | **0.87** | 38 |
| Storbreen | **0.77** | **0.8** | **0.77** | **0.79** | **0.84** | **0.87** | **0.83** | **0.85** | 30 |
| Svelgjabreen | **0.97** | **0.95** | **0.97** | **0.95** | **0.98** | **0.94** | **0.98** | **0.96** | 10 |
| Abramov glacier | **0.85** | 0.49 | **0.84** | 0.49 | **0.79** | 0.43 | **0.79** | 0.52 | 14 |
| Ts. Tuyuksuyskiy glacier | **0.43** | 0.25 | **0.47** | 0.28 | 0.35 | 0.17 | 0.26 | **0.52** | 31 |

Significant correlation (p-value < 0.10), **highly significant correlation (p-value < 0.05)**





**Table 4.** Regional means, standard deviations (std) and trend of the total precipitation over the accumulation season (original ERA-5 and MERRA-2), SWE (site-independent and season-independent GBR models) and temperature (original and downscaled ERA-5 and MERRA-2) between 1981 and 2021.

| Precipitation; SWE | | Original | | Site-indep GBR | | Season-indep GBR | |
|---|---|---|---|---|---|---|---|
| | | ERA-5 | MERRA-2 | ERA-5 | MERRA-2 | ERA-5 | MERRA-2 |
| Mean | Alps | 767 | 540 | 1160 | 1106 | 1159 | 1128 |
| [mm] | Scandinavia | 1157 | 1168 | 1964 | 2000 | 1965 | 2006 |
| | Central Asia | 301 | 155 | 383 | 470 | 385 | 403 |
| | Western Canada | 1205 | 843 | 1809 | 1605 | 1783 | 1594 |
| Mean std | Alps | 184 | 106 | 306 | 254 | 319 | 287 |
| [mm] | Scandinavia | 436 | 454 | 714 | 749 | 719 | 752 |
| | Central Asia | 143 | 55 | 161 | 138 | 162 | 148 |
| | Western Canada | 452 | 373 | 455 | 516 | 441 | 530 |
| Trend | Alps | 0.47 (0.1) | 1.65 (0.3) | 1.82 (0.2) | 3.23 (0.3) | 2.21 (0.2) | 3.95 (0.4) |
| [mm/year] | Scandinavia | 0.82 (0.1) | -1.62 (-0.1) | -4.11 (-0.2) | -11.23 (-0.6) | -5.37 (-0.3) | -9.17 (-0.5) |
| ([%/year]) | Central Asia | 0.22 (0.1) | 0.69 (0.4) | -0.34 (-0.1) | 0.66 (0.1) | -0.77 (-0.2) | -1.36 (-0.3) |
| | Western Canada | 1.61 (0.1) | 1.1 (0.1) | 4.06 (0.2) | 2.27 (0.1) | 4.10 (0.2) | 2.12 (0.1) |

| Temperature | | Original | | Downscaled | |
|---|---|---|---|---|---|
| | | ERA-5 | MERRA-2 | ERA-5 | MERRA-2 |
| Mean | Alps | -4.79 | -3.19 | -6.62 | -5.95 |
| [°C] | Scandinavia | -4.28 | -4.84 | -5.68 | -5.13 |
| | Central Asia | -12.97 | -9.58 | -13.11 | -12.31 |
| | Western Canada | -5.92 | -7.02 | -6.53 | -6.19 |
| Mean std | Alps | 1.64 | 1.49 | 1.12 | 1.04 |
| [°C] | Scandinavia | 2.52 | 3.04 | 1.99 | 1.76 |
| | Central Asia | 3.6 | 3.21 | 1.46 | 1.51 |
| | Western Canada | 1.96 | 1.68 | 1.33 | 1.26 |
| Trend | Alps | **0.06** | **0.04** | **0.03** | **0.03** |
| [°C/year] | Scandinavia | **0.06** | 0.02 | **0.04** | **0.03** |
| | Central Asia | 0.00 | -0.01 | 0.01 | 0.00 |
| | Western Canada | 0.02 | 0.00 | 0.00 | 0.00 |

Significant trends (p-value < 0.10), **highly significant trends (p-value < 0.05)**





**Table B1.** ERA-5 variables used in the study.

| Product type | Variable abbreviation | Variable full name |
|---|---|---|
| ERA-5 constants | z | Surface geopotential |
| | anor | Angle of sub-gridscale orography |
| | isor | Anisotrpy of sub-gridscale orography |
| | slor | Slope of sub-gridscale orography |
| | sdor | Standard deviation of orography |
| ERA-5 single levels | u100, u10 | 100, 10 m U wind component |
| | v100, v10 | 100, 10 m V wind component |
| | d2m | 2 m dew point temperature |
| | t2m | 2 m temperature |
| | bld | Boundary layer dissipation |
| | blh | Boundary layer height |
| | cp | Convective precipitation |
| | csf | Convective snowfall |
| | lsp | Large-scale precipitation |
| | lspf | Large-scale precipitation fraction |
| | lsf | Large-scale snowfall |
| | msl | Mean sea level pressure |
| | sf | Snowfall |
| | slhf | Surface latent heat flux |
| | ssr | Surface net solar radiation |
| | str | Surface net thermal radiation |
| | sp | Surface pressure |
| | sshf | Surface sensible heat flux |
| | tcrw | Total column rain water |
| | tcsw | Total column snow water |
| | tp* | Total precipitation |
| | p54.162 | Vertical integral of temperature |
| | deg0l | 0 degrees C isothermal level |
| ERA-5 pressure levels | t | Temperature |
| at 1000, 850, 700, 500, 400, 300 hPa | r | Relative humidity |
| | w | Vertical velocity |

*tp was used as precipitation variable.



**Table B2.** MERRA-2 variables used in the study.

| Product type | Variable abbreviation | Variable full name |
|---|---|---|
| MERRA-2 constants | PHYS | Surface geopotential height |
| | SGH | Isotropic stdv of GWD topography |
| MERRA-2 land surface diagnostics | PRECSNOLAND | Snowfall |
| | PRECTOTLAND* | Total precipitation |
| | TSURF | Surface temperature |
| MERRA-2 single-level diagnostics | CLDPRS | Cloud top pressure |
| | CLDTMP | Cloud top temperature |
| | DISPH | Zero plane displacement height |
| | H100, H850, H500, H250 | Height at 1000, 850, 500, 250 mb |
| | OMEGA500 | Omega at 500 hPa |
| | PBLTOP | Pbltop pressure |
| | PS | Surface pressure |
| | Q850, Q500, Q250 | Specific humidity at 850, 500, 250 hPa |
| | QV10M, QV2M | 10, 2 m specific humidity |
| | SLP | Sea level pressure |
| | T10M, T2M | 10, 2 m air temperature |
| | T850, T500, T250 | Air temperature at 850, 500, 250 hPa |
| | T2MDEW | Dew point temperature at 2 m |
| | T2MWET | Wet bulb temperature at 2 m |
| | TQI | Total precipitable ice water |
| | TQL | Total precipitable liquid water |
| | TQV | Total precipitable water vapour |
| | TROPPB | Tropopause pressure, blended estimate |
| | TROPPT | Tropopause pressure, thermal estimate |
| | TROPPV | Tropopause pressure, EPV estimate |
| | TROPQ | Tropopause specific humidity, blended estimate |
| | TROPT | Tropopause temperature, blended estimate |
| | U50M, U10M, U2M | 50, 10, 2 m eastward wind |
| | U850, U500, U250 | Eastward wind at 850, 500, 250 hPa |
| | V50M, V10M, V2M | 50, 10, 2 m northward wind |
| | V850, V500, V250 | Northward wind at 850, 500, 250 hPa |
| MERRA-2 analyzed meteorological fields at 1000 to | T | Air emperature |
| 700 hPa (25 hPa steps) and 700 to 400 hPa (50 hPa steps) | QV | Specific humidity |

*PRECTOTLAND was used as precipitation variable.



**Table B3.** Variables used by the GBR models. The name "variable_down" refers to the variable downscaled at the elevation of the $B_w$ data (linear interpolation from the pressure levels data), "variable_Ppos" refers to the mean of the variable during the accumulation season considering only days with a minimum precipitation of 5 mm, "delta_variable" refers to the difference between "variable_down" and the variable at the original grid of the reanalysis. $P_{grid}$ is the precipitation at the grid of the reanalysis. The variable names have the same roots as those reported in Tables B1 and B2.

| GBR model | Static or seasonal | Original | Downscaled | delta_variable | variable_Ppos (Means with $P_{grid} > 5mm$) |
|---|---|---|---|---|---|
| ERA-5 | H_obs, H_grid, lat, lon, slope, cos(aspect), sin(aspect), see constants in Table B1, year | r2m*, Wh10M**, Wh100M**, see single-level in Table B1 | w_down, t_down, r_down | H_obs−H_grid t_down−t2m r_down−r2m* | w_down_Ppos, t2m_Ppos, t_down_Ppos, delta_t_Ppos, r2m_Ppos*, r_down_Ppos*, delta_r_Ppos* |
| MERRA-2 | H_obs, lat, lon, slope, cos(aspect), sin(aspect) see constants in Table B2, year | Td10M*, RH10M*, Wh10M**, Wh250**, Wh2M**, Wh500**, Wh50M**, Wh850***, see land surface and single-level in Table B2 | T_down, Td_down*, QV_down, RH_down* | H_obs−H_grid T_down−T10M Td_down−T2MDEWM QV_down−QV10M RH_down*−RH10M* | T_down_Ppos, T10M_Ppos, delta_T_Ppos Td_down_Ppos, delta_Td_Ppos QV_down_Ppos, QV10M_Ppos, delta_QV_Ppos RH_down_Ppos*, RH10M_Ppos*, delta_RH_Ppos* |

*These variables were derived using the equations reported in Section A. **These variables represent the wind speed derived with the two horizontal components.