# Peer review of "Spatio-temporal reconstruction of winter glacier mass balance in the Alps, Scandinavia, Central Asia and Western Canada (1981-2019) using climate reanalyses and machine learning"

_The Cryosphere, 2022_

## Author Response (AR1)

**Response letter**

**Snow accumulation over glaciers in the Alps, Scandinavia, Central Asia and Western Canada (1981-2021) inferred from climate reanalyses and machine learning**

Guidicelli, M., Huss, M., Gabella, M., and Salzmann, N.

Dear Editor,

Many thanks for your valuable work for the sake of science. We are also grateful to the reviewers' time and efforts to help improving our study.

In response to the two reviews we have thoroughly and substantially revised our manuscript. Below, please find a brief summary of the most important changes of the study (incl. substantial re-calculations, adjustments of the analyses schemes and a new sensitivity study) and related changes in the manuscript:

- The manuscript has been shortened by removing the SWE-trends analysis. Thus, the general goal, the abstract, the introduction and the conclusion have been rewritten by focusing more on the gradient boosting regressor (GBR) models and their ability of adjusting the reanalysis.
- The cross-validation and test schemes have been modified. Each glacier, in turn, is used to test the model trained and validated with the other glaciers. All the results concerning the GBR models have thus slightly changed. However, none of the changes affected our discussion and the conclusion.
- The analysis of the predictors' importance has been performed differently. The overall GBR models' performance in terms of root-mean-squared-error (against the snow accumulation data) is evaluated depending on different groups of predictors.
- The spatial generalization capability of the models is now discussed in a more critical way.
- The ability of the models to represent the temporal variability of the snow accumulation has also been more critically discussed and a new sensitivity study is reported in the supplement.

We are confident that these changes have significantly improved our work so that it finally will reach the needed quality and standard to be accepted and published in 'The Cryosphere'.

On behalf of all co-authors

Matteo Guidicelli
* * *
**Comments by the Editor**

Legend: Editor's comments; *authors replies*
* * *
Manuscript excerpts with added text and
* * ** * *
Some minor comments after my overview read are:

(a) when you review the state of the art in the introduction, I think it would be good to at least mention dynamical downscaling, which has been employed more and more over glacier regions in recent years. A relevant paper could be Mölg & Kaser (2011, JGR Atmospheres, 116: D16101). I want to emphasize that I am not suggesting this source because I am an author on it, but because if it was one of the first (or the first) study to evaluate high-res dynamical downscaling against measurements taken on a glacier (including precipitation). Hence the study could be a relevant addition to the refs:

*Dynamical downscaling and Mölg & Kaser (2011, JGR Atmospheres, 116: D16101) are now mentioned in the introduction at lines 58-60:*

 As a result, downscaling of precipitation estimates of reanalyses is necessary to represent the local conditions in high-mountain regions. Different statistical and dynamical downscaling methods exist (cf. Maraun et al., 2010), which has also been employed and evaluated over glacierized regions (e.g. Mölg and Kaser, 2011).

(b) The very large tables 2 and 3 could be moved to the supplement:

*Tab. 2 was moved to the supplement, while tab. 3 was shortened by including only glaciers with a record of at least 15 season of snow accumulation data. This table was kept in the manuscript because it is needed to support the analysis of the models' performance to represent the temporal variability of the snow accumulation.*

(c) I am also saying this because I would avoid having a supplement for only one figure:

*The supplement is now composed by three figures and a table.*
* * *
**Comments by Reviewer 1**

Legend: Reviewer's comments; *authors replies*

| Manuscript excerpts with added text and  |
|---|
* * *
*We would like to acknowledge the reviewer for this thorough and critical review that has helped us to sharpen the focus of our study.*

The problems begin with the title, which overstates its importance. Only a tiny fraction, in fewer than half of the continents, of the world's glaciers are examined. The manuscript has too many figures and tables. The manuscript is supposed to be within 12 journal pages for TCD. The tables and figures alone, most of which occupy a full page, would take up this much space. The figures are bloated. For example, there is no need to illustrate "Tree 1" nor "Tree N", both of which are identical in Figure 3. The PCA section (4.1) doesn't tell the reader much more than the fact that elevation is the most important downscaling predictor:

*Title:*
*We agree that the term "world's glaciers" can be misleading. In response to this comment, we changed the title to: "Snow accumulation over glaciers in the Alps, Scandinavia, Central Asia and Western Canada (1981-2020) inferred from climate reanalysis and machine learning"*

**Number of figures and tables**:
*We agree that some simplification was beneficial to the paper and we accordingly performed major changes including a reduction of the number of Figures / Tables:*

- *The manuscript was shortened by removing the SWE-trends analysis (old Sec. 3.4, 4.3 and 5.2; Tab. 4; Fig. 11 and 12 were removed). Thus, the general goal, the introduction and the conclusion were modified by focusing more on the GBR models and their ability of adjusting the reanalysis.*
- *Tab. 2 was moved to the supplement.*
- *Tab. 3 (new Tab. 2) was simplified by including only glaciers with a record of at least 15 season of snow accumulation data. This table was kept in the manuscript because it is needed to support the analysis of the models' performance to represent the temporal variability of the snow accumulation.*
- *Fig. 2 was moved to the Supplement.*
- *Fig. 3 was simplified and replaced by a smaller figure without the illustration of the "Trees" (new Fig. 2), which only described the cross-validation and test schemes.*
- *We also agree that Sec. 4.1 needed to be rewritten in order to better quantify the added value of each group of predictors on the model's performance. Thus, Fig. 4 was removed and Fig. 5 was moved to the supplement as it shows that other predictors than elevation are important to explain different biases between reanalysis' precipitation and snow accumulation on glaciers. In the revised version of the paper, we showed the changes in terms of overall model performance when suppressing groups of predictors (new Fig. 3).*

*The results of the new Fig. 3 are described at the beginning of Sec. 4.1:*

| In order to understand the importance of the predictors used by the GBR models (i.e. those not related to the elevation of the glaciers and their elevation difference with the reanalysis' grid), we evaluated the changes in terms of overall GBR model performance when suppressing groups of predictors. For both ERA-5 and MERRA-2 site-independent GBR models, the smallest RMSE results when using all predictors (Fig. 3a and b). The RMSE particularly increases when suppressing the MERRA-2 single level and pressure levels variables from the predictors. In turn, for both ERA-5 and MERRA-2 season-independent GBR models, the smallest RMSE results when suppressing the single level and pressure |
|---|

levels variables from the predictors (Fig. 3c and d). The RMSE increases most when suppressing the year, the topographical parameters and the glaciers coordinates simultaneously as predictors.

However, skipping reanalysis variables from the set of predictors leads to higher errors for some individual glaciers, especially in the representation of the temporal variability of the snow accumulation. In fact, excluding the reanalysis variables, the year is the only predictor that could be used to predict a different adjustment factor depending on the accumulation season (all the other predictors are constant in time). Therefore, and to allow a fairer comparison between site-independent and season-independent GBRs, in all our following analyses we always include all predictors.

The leave one out validation is problematic as there is no independent validation dataset used, meaning that biases in precipitation are unlikely to be identified:

*Many thanks for this thought. However, we do not fully agree with this statement. For the "site-independent GBR", the model was always validated on a glacier that was independent from the model's training. Thus, as stated in the manuscript, the leave-one glacier-out cross-validation allowed evaluating the generalization of the machine-learning models for glaciers located in the same regions of the training data. Fig. 7 (old Fig. 9) shows a more robust validation, where the performance of the machine-learning models is also evaluated for completely independent regions (removing neighboring glaciers from the training data). Biases of reanalysis's precipitation against snow accumulation data (based on ground measurements and extrapolation techniques (see Sec. 2.2)) on the glaciers of the study are therefore identified (see Fig. 4 and 5 (old Fig. 6 and 7)).*

*Despite the glaciers used for validation being independent from the GBR model's training, it is true that they had an influence on the choice of the optimal hyperparameters of the GBR model, i.e.: the GBR model was optimized to perform well on the validation data. However, each single glacier (1 out of 95 glaciers) used for the validation had a very limited weight on the overall performance (mean squared error) and on the choice of the GBR's hyperparameters.*

*In order to make the proposed methodology even more robust, we also defined the hyperparameters independently from the test sites, i.e.: in turn, each glacier was used to test the GBR model trained and validated (10-fold cross-validation for the selection of the hyperparameters) with the other glaciers. As a consequence of the new training, all the results regarding the GBR models have slightly changed. However, this did not affect our discussion nor the conclusion of the study.*

*The new cross-validation and test scheme is illustrated in the new Fig. 2. and is described at lines 191-210:*

Different hyperparameters characterize a GBR. In this study, we applied a grid search to optimize the number of estimators (number of additive trees), the maximum depth that each tree can reach, the minimum number of samples required to be at a leaf node of a tree, and the  learning rate. A 10-fold cross-validation was applied with different combinations of hyperparameters. The hyperparameters that were able to minimize the mean squared error of the validation data were chosen.  Finally, the GBR model with the chosen hyperparameters was tested on independent data.

The validation  and the test data were defined differently depending on the goal of the GBR model. For both reanalysis products (ERA-5 and MERRA-2), we built two different GBR models with two different goals and two different cross-validation  and test schemes. The first GBR model is site-independent and aims at "extrapolating" the Bw data in time and space (over glaciers with no Bw data).  Thus, groups of data (folds) in the 10-fold cross-validation contain data of different glaciers and the site-independent GBR model with the chosen hyperparameters was tested on an independent glacier. This process was repeated for each glacier, which was used to test the GBR-model defined with the data of the other glaciers (see Fig. 2). The second GBR model is season independent and aims at "extrapolating" the Bw data in time only (filling data gaps over glaciers with discontinuous records of

Bw). For these cases,  groups of data in the 10-fold cross-validation  contain data of different years but different groups can contain data of different years of the same glacier. Finally, the season-independent GBR model with the chosen hyperparameters was tested for an independent year of a given glacier. This process was repeated for each year and each glacier.

The average optimal hyperparameters for all the studied glaciers are reported in Tab. 1. The resulting site-independent model is more generalized (since no information regarding the glacier where the model is validated and tested was provided), while the season-independent model is more detailed and  can split into individual sub-models adapted to a small number of samples (since it can exploit the Bw data  of the tested glacier).

ERA-5 and MERRA-2 reanalyses are used without any mention of their potential large biases in the mountains. For example, Liu and Margulis (2019) report that MERRA-2 underestimates snowfall (which is based on the "PRECTTOLAND" variable used here) by 54% in High Mountain Asia.:

*We are fully aware of the limitations of Reanalyses (because of missing and/or highly inaccurate in-situ observations) in high mountain region and specifically precipitation. In fact, our whole study is in principle motivated by this major challenge of improving the quantification of high altitude (solid) precipitation and SWE. In the manuscript, reanalysis biases in high-mountain regions were thus mentioned including references in the introduction. However, we agree that the biases observed in previous studies have not been described and quantified abundantly enough. In the revised paper we better included them in the introduction thus enhancing the comprehensiveness of the manuscript. We also added respective reference in the revised manuscript (lines 53-57).*

However, the performance of reanalysis results can vary greatly depending on the region and the elevation range of interest (Sun et al., 2018). Large biases in reanalysis precipitation are particularly observed in high-mountain regions (e.g. Liu and Margulis, 2019; Zandler et al., 2019) . The scarcity of observations  available for assimilation and the coarse resolution of such models limit their accuracy in areas of complex topography and their suitability for studies at a local scale (e.g. Salzmann and Mearns, 2012, (for snow)).

*New cited references:*
- *Zandler, H., Haag, I., and Samimi, C.: Evaluation needs and temporal performance differences of gridded precipitation products in peripheral mountain regions, Scientific Reports, 9, 15 118, https://doi.org/10.1038/s41598-019-51666-z, 2019.*
- *Liu, Y. and Margulis, S. A.: Deriving Bias and Uncertainty in MERRA-2 Snowfall Precipitation Over High Mountain Asia, Frontiers in Earth Science, 7, 280, https://doi.org/10.3389/feart.2019.00280, 2019.*

It's not clear to me that the downscaling techniques presented here will correct that bias, as no independent evaluation of precipitation is presented:

*Reanalysis's precipitation is compared against snow accumulation data on glaciers. This data clearly is independent, and it is to our knowledge the only and thus best possible source of (cumulative) precipitation at very high elevation. The machine-learning model is trained, validated and tested against these snow accumulation data on glaciers. In general, from the results presented in the revised manuscript (e.g. Figs. 4 and 5) it is clear that, on average, the machine-learning models can adjust the reanalysis' bias against snow accumulation on glaciers, which is among the main purposes of the study. We hope that this response answers the reviewer's comment. Otherwise, we would be happy to obtain additional explanations.*

Melt and sublimation are ignored in the "winter mass balance," which is then the wrong term:

*We do not fully agree with the reviewer here. The term "winter mass balance" refers to the snow water equivalent found on the glacier close to the maximum of snow depth, or the end of winter. Therefore, the winter mass balance – per definition – includes loss terms such as melt and sublimation, although they are not individually quantified. Furthermore, our periods of analysis are adjusted to optimally match the period where the components of melt and sublimation are small in comparison to accumulation by solid precipitation.*

After carefully searching through the text, I still cannot understand how precipitation phase was treated. It seems to have been ignored as SWE is used interchangeably with the downscaled precipitation on glaciers. But then, in Table B1 and B2 ERA-5/MERRA-2 snowfall variables are listed as predictors?:

*Indeed, the precipitation phase was ignored. In the revised paper (lines 144-152) we more clearly described this choice and the reason of including the snowfall variable in the predictors:*

First, we derived total or average of all variables provided by the reanalyses for the entire accumulation season. Subsequently, a machine learning model to adjust the total precipitation (see Sec. 2.1.1 and 2.1.2) of the reanalyses over glaciers for the accumulation season was developed to derive SWE estimates. We use a GBR (gradient boosting regressor), which makes use of several meteorological variables (original and downscaled) and topographical parameters as input variables (predictors). In principle, a different adjustment factor of precipitation might be needed depending on the precipitation phase. However, as we only adjust the total precipitation occurring during the accumulation season, the adjustment factors used here represent the "average" adjustment factor of all precipitation events. Moreover, the snowfall variable was used as a predictor in order to enable the GBR model to learn that a different "average" adjustment factor must be applied depending on the fraction of snowfall and total precipitation (i.e. depending on the main precipitation phase during the accumulation season).
* * *
**Comments by Reviewer 2**

Legend: Reviewer's comments; *authors replies*

Manuscript excerpts with added text and
* * *
Guidicelli et al propose an interesting method to downscale and bias-correct reanalysis precipitation data to the elevation and sites of glaciers in 4 regions of the world. 2 reanalyses are used : ERA5 and MERRA2. The method is based on gradient boosting regressions, a technique from the field of artificial intelligence. The performance of this method is evaluated through cross-validation and discussed in terms of both temporal and spatial extrapolation. Finally, precipitation trends on glaciers are derived for each 4 regions based on the bias corrected and downscaled reanalysis data.

The study tackles the very interesting and yet unsolved issue of high-altitude precipitation amounts, with tools from machine learning. It adds to the existing literature by focusing on glacier winter mass balances, used as a proxy for winter precipitation at high altitudes. In my opinion, this makes the topic of this study very relevant. While the analyses displayed are in general sound, I advise a revision of the paper with respect to concerns regarding the spatial generalization capability of the models and the derivation of trends, see below.

*We would like to thank the reviewer for the positive appreciation of our work and the constructive comments that have helped us to improve the paper considerably.*

MAIN COMMENTS

**1 - Comparison/justification with respect to other AI techniques for bias correction and downscaling in literature :** Even though the introduction describes well the existing literature on AI-based downscaling/bias correction methods, the choice of GBR is barely justified with respect to other techniques. I would have expected elements in that direction in the manuscript, especially since a section of the Discussion is entitled : '5.1 Advantages and disadvantages of gradient boosting regressors':

*The discussion section was reorganized and a subsection (5.1.2) was dedicated to explain our choice of applying a gradient boosting regressor model (lines 378-385):*

5.1.2 Differences with other machine learning algorithms

We have chosen a tree-based algorithm because of its higher readability in terms of the predictors' usage compared to other machine learning methods (e.g. Huysmans et al., 2011; Freitas, 2014). A disadvantage of tree-based algorithms, however, could be that this approach does not predict continuous values. Yet here, we aim at predicting an adjustment factor depending on a classification based on the used predictors, which is exactly the purpose of a tree-based algorithm. The choice of a gradient boosting instead of other tree-based algorithms (e.g. random forest (Breiman, 2001)) is motivated by the fact that gradient boosting is a gradient descent algorithm, where each additional tree tries to reduce the bias (which is the main goal of our study) rather than the variance of the predictions.

*New cited references:*
- *Breiman, L.: Random Forests, Machine Learning, 45, 5–32, https://doi.org/10.1023/A:1010933404324, 2001.*
- *Freitas, A.: Comprehensible classification models: a position paper, ACM SIGKDD explorations newsletter, 15(1), 1–10, 2014.*
- *Friedman, J. H.: Greedy function approximation: a gradient boosting machine, Annals of statistics, pp. 1189–1232, 2001. (this paper is cited in the introduction (line ?))*
- *Huysmans, J., Dejaeger, K., Mues, C., Vanthienen, J., and Baesens, B.: An empirical evaluation of the comprehensibility of decision table, tree and rule based predictive models, Decision Support Systems, 51(1), 141–154, https://doi.org/10.1016/j.dss.2010.12.003, 2011.*

**2 - Limits inherent to the number of available learning data :**

Some of the regions of interest, e.g. Canada and Central Asia, have in total less than 20 glaciers used in this study, which is an extremely low percentage of the number of glaciers that they truly host.

This in my opinion strongly impedes the (spatial) generalization capability of the GBR models learned on these data, to the region of interest as a whole. Although this is not what the authors do in the paper, this is what the title suggests while mentioning the world's glaciers. I would strongly recommend to modify this misleading title, as the developed technique is in practice not applied to derive precipitation data over any glacier of the world, but is limited to (i) the regions of interest and (ii) the few glaciers with data in these regions.. On top of the low sampling level for application of machine learning techniques in general, there may be furthermore a strong sampling bias in the glaciers data from WGMS, for instance towards large glaciers in the European Alps, so that the representativity of the glaciers with data w/r to the regions of interest is questionable. It follows that it is hard to know whether models or conclusions inferred solely based on these very few glaciers, are representative of the region as a whole.

I very much would like the authors to comment on this.

"The good performance of the GBRs in terms of bias suggests that they can be used for SWE estimates over glaciers where no ground observations are available (site-independent GBRs)". Despite being better than the benchmark, the performance of site-independant GBR models is limited (Fig 9) and decreases when data of neighbouring glaciers are excluded from the training. Considering that, and the likely sampling biases of WGMS data, I think the authors could revise this sentence:

*We agree with the reviewer regarding most aspects mentioned here. In the revised paper we discussed more critically our approaches and also demonstrated the limitations of our approach, for example in the case of a limited number of observations.*

*Title: We agree that the term "world's glaciers" can be misleading. We changed the title to: "Snow accumulation over glaciers in the Alps, Scandinavia, Central Asia and Western Canada (1981-2020) inferred from climate reanalysis and machine learning"*

*Regarding the sentence mentioned ("The good performance of the GBRs in terms of bias suggests that they can be used for SWE estimates over glaciers where no ground observations are available (site-independent GBRs)") we fully agree that our statement was too optimistic / too general. This was better specified at lines 317-319:*

However, the performance generally decreases when the glacier is not in proximity to the glaciers used to train the GBR models. Furthermore, we assume that the resulting performance strongly depends on the characteristics of the glacier with respect to the glaciers used in the training.

*and in the discussion at lines 375-377:*

Both, the GBR models and the benchmark do not require direct ground observations to be applied. However, the performance of the GBR models is influenced by the amount of data used to train the models and strongly depends on the characteristics of the glacier with respect to the glaciers used to train the models.

**"3 - Trends :**

In my opinion the derivation of trends based on the GBR modelled precipitation, should be accompanied with sensitivity tests to ascertain the robustness and uncertainties of this method. Typically, data-withdrawal techniques could be used on the longest time-series to evaluate the robustness/uncertainty of the trends derived when missing data are encountered. The distribution of the data gaps within the time-series (= for instance one missing season every two year, vs 20 years with data and nothing for the following 20 years) may also play a role, and it would be good to have an insight into this and possibly only derive trends for glaciers with a sufficient number data (seasons). The strong limitation of temporal extrapolation for some glaciers is highlighted l 350-l355, hence making a derivation of trends on these glaciers meaningless."

*Thanks a lot, this is a very valid comment and a good suggestion.*

*In the trend analysis, the GBR models were applied over 41 years for all the glaciers of the study. The Bw data was only used to train the GBR models and not to derive the trends. We used the temporal correlation (over years) between the GBR models and the Bw data as an indicator of the trends accuracy. However, as highlighted, the number of glaciers with long records of Bw data is limited and do not allow general conclusions in terms of trends. For this reason, and in order to reduce the length and sharpen the focus of the manuscript, we decided to completely remove the trend analysis from the manuscript.*

*Nevertheless, we still discuss the possibility and the limits of deriving trends with our GBR models. This was supported by a new sensitivity test as proposed by the reviewer: the temporal correlation of the season-independent GBRs with the Bw data was evaluated depending on the number of years of data of the tested glacier used to train the GBR model (similarly to Fig. 7a, c, e and g (old Fig. 9)). This sensitivity test was performed only for glaciers with more than 30 years of Bw data available and the result is reported in the supplement.*

*In the first version of the manuscript, the trends were also derived with the site-independent GBRs, which are not affected by the number of years with available Bw data (because no Bw data of the tested glacier is used for training). Tab. 2 and Fig. S3 show that the site-independent GBRs often perform better than the season-independent GBRs in terms of temporal correlation with the Bw data. This indicates that the number of available years with Bw data does not necessarily need to be high in order to accurately represent the temporal variability of the snow accumulation over the years and thus, in order to derive trends.*

*The new results are described in Sec. 4.2.3 and a new dedicated section is reported in the discussion (5.2.3):*

5.2.3 Representation of the temporal variability of the snow accumulation

All GBRs aimed at minimizing the MSE between the predicted and reference logarithmic adjustment factors (Eq. 3). The improvement of the temporal correlation between the original reanalysis and the Bw data is thus a consequence of bias-adjusted estimates over accumulation seasons rather than a primary goal of the GBRs. A sensitivity test reported in the Supplementary material (Fig. S3) suggests that the season-independent GBRs are not very sensitive to the number of years of data of the tested glacier used for training. Their performance is comparable to the site-independent GBRs (Tab. 2). Furthermore, only in a few cases the site-independent GBRs show a performance inferior to the

original reanalysis or the benchmark method (e.g. Ts. Tuyuksuyskiy glacier). These promising results suggest that our new estimates could also be used to derive SWE trends with generally higher accuracy than the original reanalyses, thus potentially providing insights on the relation between climate change and both snow accumulation and precipitation at the highest elevations of mountain ranges, where virtually no direct precipitation records are available. Still, the limited number of glaciers with abundant Bw data coverage available over sufficient number of years do not allow us to perform a complete application of this approach.

MINOR COMMENTS

- the GBR consider as predictors both elevation differences between reanalysis pixel and glacier site, and downscaled variables like temperature, whereby the downscaling of temperature itself mostly relies on this altitude difference. Hence there is a high redundancy in the chosen predictors. Did you test suppressing the downscaled predictors ?

*Thanks for this interesting comment. The high correlation between predictors is a problem for the interpretability of the predictors' importance. However, this does not importantly affect the performance of the GBR because decision trees are by nature not affected by multi-collinearity. If two predictors are highly correlated, the tree will choose only one of the two predictors when deciding upon a split.*

*However, we agree that Sec. 4.1 needed to be modified in order to better quantify the added value of each group of predictors on the model's performance. Thus, Fig. 4 was removed and Fig. 5 was moved to the supplement. In the revised version of the paper we showed the changes in terms of overall model performance when suppressing groups of predictors (new Fig. 3). The results of the new Fig. 3 are described at the beginning of Sec. 4.1:*

In order to understand the importance of the predictors used by the GBR models (i.e. those not related to the elevation of the glaciers and their elevation difference with the reanalysis' grid), we evaluated the changes in terms of overall GBR model performance when suppressing groups of predictors. For both ERA-5 and MERRA-2 site-independent GBR models, the smallest RMSE results when using all predictors (Fig. 3a and b). The RMSE particularly increases when suppressing the MERRA-2 single level and pressure levels variables from the predictors. In turn, for both ERA-5 and MERRA-2 season-independent GBR models, the smallest RMSE results when suppressing the single level and pressure levels variables from the predictors (Fig. 3c and d). The RMSE increases most when suppressing the year, the topographical parameters and the glaciers coordinates simultaneously as predictors.

However, skipping reanalysis variables from the set of predictors leads to higher errors for some individual glaciers, especially in the representation of the temporal variability of the snow accumulation. In fact, excluding the reanalysis variables, the year is the only predictor that could be used to predict a different adjustment factor depending on the accumulation season (all the other predictors are constant in time). Therefore, and to allow a fairer comparison between site-independent and season-independent GBRs, in all our following analyses we always include all predictors.

- the predictors in the PCA figures (4 and 5) are often barely lisible. Fig 5 could maybe join the supplemental material.

*Fig. 5 was moved to the supplement. We also increased the fontsize and avoided the overlapping of predictors' names.*

- l 264-274 : could the different magnitude in factors relate to known biases / weaknesses of the reanalyses in representing different types of precipitation events ?

*Yes, thanks for this good suggestion. However, so far we were not able to directly relate our findings to statements reported in the literature.*

- l 311 : "their performance is worse than the site-independent models". It is not so clear for me why : could you please explain ?

*The season-independent GBR model has a higher number of trees and less samples are needed to create a new leaf of the tree (i.e. to predict a different adjustment factor) than the site-independent GBR. Thanks to its higher complexity than the site-independent model, if Bw data of the validated glacier is used to train the season-independent model, this latter can learn the specific characteristics of the tested glacier and perform better than the site-independent model. On the other hand, if no Bw data of the tested glacier is used to train the season-independent GBR, its performance is worse than the site-independent GBR, because it will overfit the training data.*

*A new dedicated section is reported in the discussion:*
* * *
5.2.1 Site-independent and season-independent GBRs

The lower generalization of the season-independent GBRs compared with the site-independent GBRs allows the splitting into individual sub-models adapted to a small number of samples (see Tab. 1). This enables to exploit the Bw data of the tested glacier by creating a specific sub-model, but can result in an overfit of the training data. In contrary, the higher generalization of the site-independent GBRs allows learning on overall relationships between the used predictors and the reference adjustment factors (Eq. 2).

The used training data and the selected hyperparameters also have a direct influence on the predictors needed by the GBR models to reduce the cost function (Eq. 3). In fact, the use of reanalysis variables (from single level and pressure levels) as predictors, caused an increase of the overall RMSE of both ERA-5 and MERRA-2 season-independent GBRs against the Bw data of all glaciers of the study (Fig. 3c and d). However, despite the high correlation of the downscaled reanalysis variables (cf. Section 3.1) with the elevation of the glaciers, their inclusion in the set of predictors for the training of the site-independent GBRs reduced the overall RMSE (Fig. 3a and b). This difference can be explained by the combined effect of using data of the tested glacier in the training of the season-independent GBRs, and defining a small minimum number of samples required to create a leaf node of the GBR. In fact, the season-independent GBR can theoretically exploit the coordinates to split into individual sub-models adapted to individual glaciers. Therefore, the season-independent GBRs can learn the adjustment factors observed in the other accumulation seasons of the tested glacier and predict a similar adjustment factor for the tested accumulation season, with no need of learning overall relationships between the reanalysis predictors and the reference adjustment factors (Eq. 2).
* * *
- l 448 : why were more topographic predictors used in the ERA-5 GBRs than in the MERRA-2 ones ?

*We used all the topographical predictors describing the reanalysis's subgrid complexity of both reanalysis products and ERA-5 is providing more descriptors than MERRA-2. This is now specified at lines 450-453:*
* * *
The differences between the performance of our GBR models are also caused by the different predictors that have been used. For instance,  we considered all the topographical predictors describing the  reanalysis's subgrid complexity of both reanalysis products and ERA-5 is providing more descriptors than MERRA-2  (see Tab. B1 and B2).
* * *
- Fig 2 could join the Supplemental material

*Yes, we agree. Fig.2 was moved to the supplement.*

- Fig 6 : could the absolute biases also be mentioned ?

*Yes, we also evaluated and reported the mean absolute error in addition to the root mean squared error. However, the figure is already too busy to allow more numbers and we reported the results in the text (lines 273-283):*

Figure 4 shows the comparison between all glacier-wide Bw values and the models' estimates. MERRA-2 precipitation underestimates Bw more importantly than ERA-5 precipitation in all regions (Fig. 4a and b), with an overall RMSE of 946 mm (mean absolute error (MAE) of 749 mm) against 793 mm (611 mm) of ERA-5. Excluding the Alps, the correlation between the Bw data and the ERA-5 precipitation is always higher than the correlation with the MERRA-2 precipitation. The adjusted estimates obtained with the site-independent and the season-independent GBRs allowed us to consistently reduce (increase) the bias (correlation (r)) between the precipitation of the original reanalyses and Bw (from an overall RMSE (CORR) of 946 overall RMSE of 433 mm(0.74) and 793, MAE of 326 mm(0.81) of , r of 0.86 for the:MERRA-2 and ERA-5, to 443 site-independent GBR; RMSE of 410 mm(0.85) and 422, MAE of 307 mm(0.86) of the , r of 0.87 for the ERA-5 site-independent GBRs, and 287 GBR RMSE of 293 mm MAE of 211 mm( , r of 0.94 ) and 272 for the MERRA-2 season-independent GBR RMSE of 275 mm(0.95) of the MAE of 200 mm, r of 0.94 for the ERA-5 season-independent GBRs GBR). These results demonstrate the need of an adjustment of reanalyses data to reproduce snow accumulation on glaciers, which are, otherwise, largely underestimated in all four regions involved in this study.

- Fig 7: a ranking of the glaciers with respect to altitude, or to the number of seasons with Bw_data, would enable to more efficiently support the analysis related to this figure, please consider this. The same applies to Fig 11.

*Thanks for the suggestion. We modified Fig. 7 (new Fig. 5) by ranking the glaciers with respect to the number of seasons with Bw data . Fig. 11 was removed.*

- Tables 1 and 2 could join the supplemental material

*Tab.2 was moved to the supplement. In addition, the number of glaciers reported in Tab. 3 (new Tab. 2) was reduced. Tab.1 was kept in order to show the differences in terms of hyperparameters (and generalization) between the site-independent and season-independent GBRs.*

- Section 5.2 : this recent literature could also be of interest : https://doi.org/10.5194/hess-24-5355-2020; https://doi.org/10.5194/essd-14-1707-2022 (update of Durand et al., 2009).

*Thanks. However, Sec. 5.2 was removed in the revised manuscript.*

---

## Referee Report (RR1)

**Referee Report on Guidicelli et al., 2022, 4-11-2022**

I thank the authors for the thorough revision of their manuscript, they have addressed the concerns that I raised.
I see 2 minor additional complements, that would likely further improve the manuscript without requiring much effort.

- 1/ the use of the downscaled reanalysis variables as predictors should be better explained in the Methods (section 3.1 for instance) and also on Fig 3.
The methods states that "**In addition to** the original variables, the GBR requires some downscaled variables of the reanalyses as predictors at the glacier
elevation,...." but when reading the legend of Table B3 it is less clear whether for instance all single-level variables from Tables B1 and B2, are used in addition to the ones mentioned in Table B3, or just some of them. Following, we do not know which of the reanalysis variables (downscaled ? all ?) are withdrawn in the tests from Fig 3.

2/ Fig 3 and its analysis : it is at first very surprising that the suppression of reanalysis data, leads to the best performance/smallest rmse for the season independant GBR, as reanalysis variables could be hypothesized to carry the "climatological" information shaping the temporal dynamics of the adjustment factors.
The authors analyse this finding in the new section "5.2.1 Site-independent and season-independent GBRs", whereby seasonal-independant GBR models are hypothesized to split into individual sub-models adapted to individual glaciers and hence better able to predict a glacier-specific (but maybe temporally constant ?) adjustment factor inferred from the training accumulation seasons.
Actually, I see a possible alternative explanation of this finding, which occurs concomitant with the important role of the year as a predictor for season-indep GBR : it may point towards the fact that the year may be the one predictor to actually vehicle the climatological information (the GBR model then learning 'by heart" which years correspond to negatively biased winter accumulations for the glaciers in this region, and which other years correspond to more balanced/less biased ones). Though such a model may perform well in the past, its use in extrapolation to future conditions (using for instance GCM/RCM data in lieu of reanalysis), is doomed to fail.
Maybe the authors could comment on this and add a note in the discussion if they find this explanation possibly relevant

Regards

---

## Author Response (AR2)

**Response letter**

**Spatio-temporal reconstruction of winter glacier mass balance in the Alps, Scandinavia, Central Asia and Western Canada (1981-2019) using climate reanalyses and machine learning**

Guidicelli, M., Huss, M., Gabella, M., and Salzmann, N.

Dear Editor,

We sincerely thank the editor and referees for taking the time to review our manuscript and providing constructive feedback to improve our manuscript.

In response to the detailed comments by the two referees we have thoroughly and substantially revised our manuscript. Below, please find a brief summary of the most important changes of the study and related changes in the manuscript:

- We carefully revised the terminology used in the manuscript (snow accumulation, SWE, winter mass balance), which is now more coherent with the methods that have been applied. Consequently, this has substantially improved the flow of the manuscript.
- We improved the clearness of the goals (winter mass balance reconstruction) and methods (derivation of vertical profiles of the winter mass balance). Substantial changes have been made to the text in the abstract, introduction and method sections.
- We discussed the role of the year as predictor in the gradient-boosting regressors (GBRs), and the related implications.
- We described the meaning of the hyperparameters defining the GBR.
- We further discussed the choice of applying a GBR instead of other machine-learning methods.

We are confident that these changes have significantly improved our work so that it finally will reach the needed quality and standard to be accepted and published in 'The Cryosphere'.

On behalf of all co-authors,

Matteo Guidicelli
* * *
**Comments by Anonymous Referee #2**

Legend: Referee's comments; *authors replies*

| |
|---|
| Manuscript excerpts with added text and <s>deleted text</s> |
* * *
I thank the authors for the thorough revision of their manuscript, they have addressed the concerns that I raised. I see 2 minor additional complements, that would likely further improve the manuscript without requiring much effort.

*We would like to thank the referee for the positive feedback and for the additional minor complements that have helped us to further improve the manuscript.*

1/ the use of the downscaled reanalysis variables as predictors should be better explained in the Methods (section 3.1 for instance) and also on Fig 3.
The methods state that "In addition to the original variables, the GBR requires some downscaled variables of the reanalyses as predictors at the glacier elevation, …" but when reading the legend of Table B3 it is less clear whether for instance all single level variables from Tables B1 and B2, are used in addition to the ones mentioned in Table B3, or just some of them. Following, we do not know which of the reanalysis variables (downscaled? all?) are withdrawn in the tests from Fig 3.

*All variables listed in Table B3 are required by the GBR. This is now indicated at the beginning of the Methods section (lines 147-148):*

| |
|---|
| The list of predictors required by the GBR is summarized in Tab. B3, while the variable names are described in Tabs. B1 (ERA-5) and B2 (MERRA-2). |

*and at the beginning of Section 3.1:*

| |
|---|
| In addition to the original variables<s>,</s> (all "constants", all "single level" and all "land surface" variables in Tabs. B1 and B2), the GBR requires some downscaled variables of the reanalyses as predictors at the glacier elevation intervals (see Tab. B3), including air temperature, dew point temperature and relative humidity (for MERRA-2 and ERA-5), vertical velocity of air motion (for ERA-5 only) and specific humidity (for MERRA-2 only). |

*Furthermore, the caption and the content of Table B3 have been modified in order to make clear that all "single-level" variables and "constants" of Tables B1 and B2 are required by the GBR. Here, we report the new caption of the table:*

| |
|---|
| Summary of the variables used by the GBR models: all constants and single-level variables from the original reanalyses (Tabs. B1 and B2), variables downscaled at the elevation of the $B_w$ data ("variable_down"), mean of the variables during the accumulation season considering only days with a minimum precipitation of 5 mm ("variable_Ppos"), difference between "variable_down" and the variable at the original grid of the reanalysis ("delta_variable"), year, glacier coordinates, slope and aspect. $P_{grid}$ is the precipitation of the original reanalysis. The variable names have the same roots as those reported in Tabs. B1 and B2. |

*The groups of predictors withdrawn in the tests of Fig. 3 (new Fig. 8) have been described in the caption, which is now consistent with Table B3. Here, we report the new caption of Fig. 3 (new Fig. 8):*

| |
|---|
| Overall root mean squared error (RMSE) between $B_w$ and GBR models using different groups of predictors, for all analyzed glaciers and years: (a) ERA-5 site-independent GBR, (b) MERRA-2 site-independent GBR, (c) ERA-5 season-independent GBR, and (d) MERRA-2 season-independent GBR. |

*Topography The "pressure levels" group refers to all variables derived from the pressure-levels data, i.e. "downscaled" variables and "delta_variable" (except for the elevation difference) in Tab. B3. The "single level" group refers to all "single-level" variables listed in Tab. B3. The "topography" group refers to the topographical parameters describing the reanalysis's subgrid complexity (all "constants" in Tabs. B1and B2) and the average slope and aspect of the glaciers by using the information provided in the Randolph Glacier Inventory version 6 (RGI Consortium, 2017). For the list of variables included in each group of predictors see Tables B1 (original ERA-5 variables), B2 (original MERRA-2 variables) and B3 (downscaled ERA-5 and MERRA-2 variables).

2/ Fig 3 and its analysis : it is at first very surprising that the suppression of reanalysis data, leads to the best performance/smallest rmse for the season independant GBR, as reanalysis variables could be hypothesized to carry the "climatological" information shaping the temporal dynamics of the adjustment factors.

The authors analyse this finding in the new section "5.2.1 Site-independent and season-independent GBRs", whereby seasonal-independant GBR models are hypothesized to split into individual submodels adapted to individual glaciers and hence better able to predict a glacier-specific (but maybe temporally constant ?) adjustment factor inferred from the training accumulation seasons. Actually, I see a possible alternative explanation of this finding, which occurs concomitant with the important role of the year as a predictor for season-indep GBR : it may point towards the fact that the year may be the one predictor to actually vehicle the climatological information (the GBR model then learning 'by heart" which years correspond to negatively biased winter accumulations for the glaciers in this region, and which other years correspond to more balanced/less biased ones).

Though such a model may perform well in the past, its use in extrapolation to future conditions (using for instance GCM/RCM data in lieu of reanalysis), is doomed to fail.

Maybe the authors could comment on this and add a note in the discussion if they find this explanation possibly relevant.

*Many thanks for the interesting interpretation of the role of the year as predictor for the GBR model. This is certainly a possible explanation of its importance and was only briefly mentioned in Sec. 4.2 (a short sentence has been added):*

However, skipping reanalysis variables from the set of predictors leads to higher errors for some individual glaciers, especially in the representation of the temporal variability of the $B_w$ data snow accumulation. In fact, excluding the reanalysis variables, the year is the only predictor able to vehiculate the climatological information, in other words, the year is the only predictor that could be used to predict a different adjustment factor depending on the accumulation season (all the other predictors are constant in time).

*We discussed the impact of the use of the year as predictor in Sec. 5.2.1 and Sec. 5.2.2.*

*In Sec. 5.2.1 (Site-independent and season-independent GBRs), we extended the interpretation of the results of Fig. 3 (new Fig. 8) by including your comment, which we think that completes our previous explanation (lines 425-435):*

In general, the year was included in the set of predictors for the GBRs because it may allow learning the climatological information and potential trends in terms of reanalysis biases against the $B_w$ data. This can be relevant in case reanalysis variables included in the predictors are not able to represent the hypothetic trends of these biases, which can exist due to the increasing availability of observational data that could have been assimilated by reanalysis models over the years. From Fig. 8, we notice that the year has a different impact on the site-independent and the season-independent GBRs. For the site-independent GBR, the withdrawal of the year from the predictors leads to a smaller increase of the overall RMSE than the withdrawal of the reanalysis variables (pressure levels and single level). In contrast, for the season-independent GBR, the best performance is reached by withdrawing the reanalysis variables. Thus, given the mentioned ability of the season-independent GBR to potentially create sub-models for specific glaciers using their coordinates, we think that the

year may be the one predictor to actually vehiculate the climatological information in this case (the GBR model learns which periods correspond to larger/smaller biases for the tested glacier).

*In Sec. 5.2.2 (Spatial and temporal transferability of the GBR models), we discussed the potential use of the season-independent GBR in the extrapolation to future conditions, related to the use of the year as predictor (lines 461-466).*

The inclusion of the year in the set of predictors is not problematic for temporal reconstruction of $B_w$ with limited gaps, but could be a limitation for extrapolation to future conditions (using for instance GCM/RCM data in lieu of reanalysis). In fact, even though the use of the year as predictor could allow identifying and modelling trends of reanalysis biases against the $B_w$ data, these trends would be limited to the training period of the GBRs. In contrast, the exclusive use of reanalysis variables as predictors could allow identifying and modelling trends of biases as function of specific climatological conditions represented by reanalysis variables.
* * *
**Comments by Referee #3: Maussion, Fabien**

Legend: Referee's comments; *authors replies*

| Manuscript excerpts with added text and  |
|---|
* * *
Dear authors,

I would like to start my review with an apology for the time it took me to write it. I am aware that endless review processes are a real strain (especially for PhD students), and some unplanned personal matters prevented me from writing sooner.

Another (less important) reason for the late review, however, is that the paper is a difficult read. I really enjoyed the study and I find that it is relevant and interesting, but as of the current version the manuscript feels like a puzzle that the readers have to solve by themselves, because a lot of relevant information is scattered across the manuscript. I also believe that the first round of reviews shaped the manuscript in a way which makes it less readable now. I think however that the manuscript can be brought in a reasonable shape with some restructuring.

*Many thanks for your detailed review, which helped us to again sharpen the focus of our study, and to make the entire study more linear – and thus easier to read –, and more robust and consistent with our goal.*

I will start with the most important point: what is the purpose of the paper? The title says "Snow accumulation over glaciers … inferred from climate reanalyses and machine learning". But, in reality, snow accumulation is never analyzed (or even plotted! I only see correction factors everywhere). What is analyzed is the capacity of a statistical model to reconstruct winter mass-balance (not snow) in space and time.

*The term snow accumulation or SWE were used under the assumption that snow melt is generally negligible for the considered period and high elevations. However, we agree that it is more correct to refer to winter mass balance than snow accumulation, especially when we discuss the output of our models. Now, this has been modified in the title and coherently throughout the entire paper (incl. abstract and introduction) to better guide the reader.*

Note that the study is well introduced: the problems stated in the introduction are real, and it IS a good idea to use winter mass-balance observations to look at biases in reanalysis data. At the end of the introduction, however, the authors state: "we thus aim at providing improved observation-independent SWE estimates at highest elevations of different mountain ranges across the Earth". But the manuscript does not provide anything like that, does it? I saw no SWE data, and I also didn't see a code & data availability section (against TC's policies, by the way: https://www.the-cryosphere.net/policies/data_policy.html).

I think that this is the main problem of the manuscript, as the reader is left wondering what the paper is about. Some themes which (I feel) are developed throughout the manuscript:
- Training a statistical model to reconstruct winter mass balance (WMB) from partial information
- What information is needed to do so successfully, and what problems are occurring when data becomes scarce (this is, in my view, the most interesting aspect of the study)
- What are probable bias in winter precipitation in reanalyses
- What are the differences between MERRA-2 and ERA5 (although, to be honest, I don't recall much discussion of this point despite the fact that having two datasets significantly clutters many figures in the manuscript).
- The WMB elevation profiles and how your model can sometimes reproduce those (to my surprise)

What is not developed in the manuscript:

- The difference between different statistical model choice (this is a bit of a weakness as it make the paper very descriptive, but is not a big issue in my opinion)
- Whether or not the method developed in this study will be used to develop regional products. This is very important because if yes, the paper needs to be a bit more careful in its wording as suggested by Reviewer #2. The paper is already much better at discussing limitations, but I think that if the plan is to derive actual products from the method, the abstract needs to state that this is the goal and to be more precise about what's needed to reach this future goal.
- If the goal is not to make some sort of product in a future paper, then I would like to suggest going back to my point above about clear study motivation statements.

This may sound like harsh comments, but I do not intend them to be that way: I think that the study has potential! It would be very beneficial to the paper to be more clearly written, to better explain what is done and why. I'll do my best to provide a more timely review at the next iteration.

*Thank you very much for your clear indications and remarks. Our replies regarding the introduction and the goals of the paper are provided by following your further specific comments below.*

*We also added the data availability section to the paper. Some contents have been moved from the acknowledgments section to the data availability section, which has been included at lines 600-608:*
* * *
Data availability.

Winter mass balance data separated per elevation intervals are freely available at https://wgms.ch/data_databaseversions/ (EE-MASS-BALANCE data sheet in WGMS, 2021, last access: 01.06.2021). The average slope and aspect of the glaciers were obtained from the Randolph Glacier Inventory version 6 at https://www.glims.org/RGI/ (last access: 05.05.2021, RGI Consortium, 2017). ERA-5 hourly data on single levels and on pressure levels were downloaded from the Copernicus Climate Change Service (C3S) Climate Data Store at https://doi.org/10.24381/cds.adbb2d47 (last access: 01.06.2021) and https://doi.org/10.24381/cds.bd0915c6 (last access: 01.06.2021), respectively (Hersbach et al., 2018b, a). MERRA-2 Land Surface Diagnostics, MERRA-2 Single-Level Diagnostics and MERRA-2 Analyzed Meteorological Fields data are available at https://doi.org/10.5067/RKPHT8KC1Y1T (last access: 13.06.2021), https://doi.org/10.5067/VJAFPLI1CSIV (last access: 13.06.2021) and https://doi.org/10.5067/A7S6XP56VZWS (last access: 13.06.2021), respectively (Global Modeling and Assimilation Office (GMAO), 2015b, c, a).
* * *
**Specific comments**

- I still don't think that the title reflects the content of the paper well (see general comments)

*The title has been changed in order to better reflect the content of the paper. The period reported in the title and in the manuscript has also been modified for coherence with the most recent winter mass balance data available from the WGMS dataset at the moment our study was conducted. In fact, the previous period (1981-2021) was related to analyses performed in the original version of the manuscript (climatological trends) but the related section has been removed after the first round of reviews.*
* * *
Spatio-temporal reconstruction of winter glacier mass balance in the Alps, Scandinavia, Central Asia and Western Canada (1981-2019) using climate reanalyses and machine learning
* * *
- Introduction: clearly state the objectives of the study, and what will be shown in the paper. Why are these regions / glaciers chosen, etc.

*Substantial changes have been made to the introduction and the abstract in order to better highlight the motivation and the contents of the study, as well as to justify the selection of the regions/glaciers analyzed in the study. We also make clear that the method developed in the study will not be used to develop regional products (in response to the general comment above), indicating that this is an exploratory study (now, this is also indicated in the abstract (line 6)). Here, we report the new last paragraph of the introduction section:*

In this study, we thus aim at  analyzing total precipitation biases of reanalysis datasets (ERA-5 and MERRA-2) over the snow accumulation season on glaciers, i.e. at the highest elevations of different mountain ranges . The precipitation estimates are compared with the winter glacier mass balance data covering a period of up to 39 years from 95 glaciers in the European Alps, Scandinavia, Central Asia and Western Canada. The selection of these regions/glaciers depends on the data consistency and availability (see Sec. 2.2). Ultimately, we aim at reconstructing the winter glacier mass balance from partial information. In order to achieve this goal, we develop and evaluate a machine-learning approach based on gradient boosting regressor (GBR) models (see Friedman, 2001) to adjust the total precipitation of reanalysis  (main driver of snow accumulation) along the elevation profiles of the glaciers. More specifically, the GBR models aim at allowing the spatiotemporal transferability of the learned information over the 95 glaciers to other glaciers with no ground observations and/or filling gaps of observational series. The new information provided by our exploratory study is expected to be helpful to improve the calibration of glaciological and hydrological models in observation-scarce regions.

- Line 144: the motivation and implication of using total seasonal averages needs to be discussed in depth. Intuitively, a model using temporal information (even at the monthly scale) would perform better, but I understand that this is not feasible in this context.

*The motivation is given by the absence of reference data at higher temporal scales (e.g. monthly) and at high elevations. The data compilation provided by the WGMS based on dozens of detailed national monitoring programs is unique in terms of providing consistent in situ data in different regions of the world at very-high elevations, i.e. on glaciers, that can be used for comparison with precipitation datasets. This comment has been added at the beginning of Sec. 2.2 (winter mass balance data, lines 106-108). We think that jointly with the reformulation of the goals of the study (where Sec. 2.2 has been cited), this is now much clearer:*

The data compilation provided by the WGMS based on dozens of detailed national monitoring programs is unique in terms of providing consistent in-situ data in different regions of the world at the highest elevation of mountain ranges, i.e. on glaciers, that can be used for comparison with precipitation datasets.

*The implication of using total seasonal averages has been discussed at the end of Sec. 5.2.2. Given the context of the study, we could only speculate that the performance would be better by using information at higher temporal scales, however, such information is not available and we could not support our comments with any data. Thus, we rather focused on the description of the models' limitations (lines 467-474):*

Furthermore, the use of total seasonal averages as inputs for the GBRs has limitations for the application of the presented approach. The GBRs provide an estimate of the overall adjustment factor for the reanalysis precipitation according to the average climatic conditions of the accumulation season. Thus, a unique adjustment factor is estimated for the whole accumulation season. The application of this adjustment factor to daily precipitation data, would allow obtaining an average estimate of the SWE evolution during the accumulation season. However, the obtained SWE evolution would neglect potential melt of snow during the accumulation season (see Sec. 2.2), as well as

potentially different adjustment factors for precipitation at higher temporal scales than seasonal. This limitation is related to the lack of reference in-situ observations at higher temporal resolution than seasonal available for our study, focusing in different regions of the world and at high elevations.

- The methods section feels incomplete. I truly don't understand how your model is actually able to simulate WMB profiles, because my understanding at the end of the methods section is that you use seasonal totals of climate predictors to simulate total seasonal WMB of glaciers. It's not clear to what purpose "downscaling" is used, and to what elevation the variables are downscaled (I assumed the average glacier elevation). Are you using elevation bands to reconstruct WMB as a function of elevation and then average per area somehow to get the glacier specific WMB? Where is this procedure described? Or, do you actually use elevation band data from WGMS for training? You can see that I'm confused.

*For each glacier we used the WGMS winter mass balance data separated per elevation intervals. All elevation intervals have been used in the training of the GBR models, which are thus able to provide estimates at different elevations for the same glacier. This information has been better highlighted in different parts of the manuscript:*
- *Abstract and Introduction: we specified that reanalysis precipitation is adjusted along the elevation profiles of the glaciers (lines 10; 76-77)*
- *Data (Sec. 2.2): we explain that we used the winter mass balance data separated per elevation intervals (lines 109; 112)*
- *Methods (first paragraph, Sec. 3.1, Sec. 3.2): we repeated both indications multiple times (lines 145; 157; 160-161; 167-168; 188)*
*We also added further information at the end of Sec. 3.2.2 (lines 229:231):*

During the training phase of our models, the $B_w$ data were weighted by considering the area of the glacier related to the respective elevation interval. Larger glaciers (and elevation intervals related to larger areas) thus receive more weight  than smaller glaciers (and elevation intervals related to smaller areas).

*However, the majority of the results shown in the study are based on glacier-wide averages as indicated in Sec. 3.3.2, where we show how the glacier-wide averages are computed.*

- L175: to be honest I don't think the benchmark is very fair, because the parameter K seems to be a parameter to tune for each reanalysis / situation. It is also not data informed at all. I am not requesting to change this at this stage, but I personally don't put much value in this benchmark.

*Thank you for this comment. We agree that the comparison between the benchmark and the GBRs is not very fair (for the season-independent GBR especially). We included the benchmark in order to show what we can expect by applying a simpler and completely not data-informed model in the case of the selected glaciers. We think that this can help the reader to understand the added value of the GBRs in terms of overall bias and vertical profiles for winter glacier mass balance. The benchmark shows that for glaciers with a non-continuously increasing $B_w$ along the profile, a more sophisticated model is needed. In order to avoid potential misinterpretations, we specified more clearly that the benchmark method is not data-informed (at the beginning of Sec. 3.2, line 168).*

*This has been also remarked in the Discussion (Sec. 5.1.1, lines 378; 390).*

- L204: " For these cases, groups of data in the 10-fold cross-validation contain data of different years but different groups can contain data of different years of the same glacier." -> this is really unclear. I assume only one glacier is used each time? Are you therefore building 95 models (one for each glacier) here? After reading the rest of the manuscript I see its not, but I really wonder what value there is to interpolate in time with a model that is trained on highly inhomogeneous data, and it seems that the data with the most explanatory power is obviously the data on this very glacier.

*We have rewritten this paragraph in order to make clearer how both site-independent and season-independent GBRs are trained and tested. Here, we report the new paragraph for readability (lines 207:217):*

The validation and the test data were defined differently depending on the goal of the GBR model. For both reanalysis products (ERA-5 and MERRA-2), we built two different GBR models with two different goals and two different cross-validation and test schemes.

The first GBR model is site-independent and aims at "extrapolating" the $B_w$ data in time and space (over glaciers with no $B_w$ data). For the site-independent GBR, we built 95 models, one for each glacier, trained and validated with the data of the other 94 glaciers. Each glacier is used, in turn, as an independent test for the model based on the data of the other 94 glaciers. Thus, the site-independent GBR is independent from any data of the glacier where the model is tested (see Fig. 2).

The second GBR model is season-independent and aims at "extrapolating" the $B_w$ data in time only (filling data gaps over glaciers with discontinuous records of $B_w$). For the season-independent GBR, we built a model for each year (when the $B_w$ data is available) of each glacier. In this case, each year of each glacier is used, in turn, as an independent test for the GBR, which is trained using the data of the other years (of the tested glacier) and of the other 94 glaciers.

*Moreover, the motivation of including also the data of the other 94 glaciers in the training of the season-independent GBR has been described at lines 217-221:*

Thus, unlike the site-independent GBR, the season-independent GBR is informed with data of the glacier where the model is tested, excluding only data of the year when the model is tested. For the season-independent GBR, data of the other 94 glaciers is still used in the training because many glaciers only have a small number of years with available $B_w$ data. Thus, in case of limited $B_w$ data, this may help the GBR to learn from the data of the other 94 glaciers.

- L206-210: this paragraph is very unclear. It's also not clear what the non-GBR specialist can learn from table 1? Either discuss to explain the value of this information or delete.

*We added an explanatory sentence at lines 199:204, where the hyperparameters are introduced:*

Different hyperparameters characterize a GBR. In this study, we applied a grid search to optimize the number of estimators (number of additive trees, i.e. number of iterations), the maximum depth that each tree can reach, the minimum number of samples required to be at a leaf node of a tree, and the learning rate. The higher the number of estimators or the maximum depth is, the more complex and less generalized the GBR model is. In contrast, the larger the minimum number of samples is, the less complex and more generalized the GBR model is. A smaller learning rate makes the GBR more robust to the specific characteristics of each individual tree, thus allowing a better generalization. However, the smaller the learning rate is, the more subsequent trees (iterations) are generally required to reach the minimum of the cost function. A 10-fold cross-validation was applied with different combinations of hyperparameters. The hyperparameters that were able to minimize the mean squared error of the validation data were chosen. Finally, the GBR model with the chosen hyperparameters was tested on independent data.

*Similar information is now reported in the caption of the table. Furthermore, we related the contents of the table with its description at lines 212:226, and we report the reader to the related discussion:*

The average optimal hyperparameters for all the studied glaciers are reported in Tab. 1. The resulting site-independent model is more generalized (smaller number of estimators than the season-independent GBR and higher minimum number of samples per leaf), while the season-independent model is more detailed and can be split into individual sub-models adapted to a small number of samples. The different architecture between the site-independent and the season-independent GBRs is discussed in Sec. 5.2.1.

- Section 4.1: intuitively, I would put this section later in the paper. But I leave this open.

*Thanks for the suggestion that we considered. We decided to move this section (new Sec. 4.2) after the main results of the study (new Sec. 4.1).*

- L250: I don't think that the calendar year should be part of the predictor pool. If a constant line has predictive power, it's because the training data have trends that are not in the reanalysis data, and I think it is highly problematic to rely on such information when trying to extrapolate in space and time. Happy to be convinced otherwise though.

*By following also the suggestion of Referee #2, we further discussed the implication of the use of the year in the set of predictors (lines 425-435). We also discussed its limitations for a potential use of the GBRs for extrapolation in the future, which indeed would be problematic (lines 461-466).*

- L253-256: this is very unclear, I'm sorry but I don't understand what this means.

*We reformulated the sentence in the middle of this paragraph, which we agree was confusing (lines 352:357):*
* * *
However, skipping reanalysis variables from the set of predictors leads to higher errors for some individual glaciers, especially in the representation of the temporal variability of the $B_w$ data . In fact, excluding the reanalysis variables, the year is the only predictor able to vehiculate the climatological information, in other words, the year is the only predictor that could be used to predict a different adjustment factor depending on the accumulation season (all the other predictors are constant in time).
* * *
- L278-281: Isn't this information already on the figure and does it need repeating here?

*In the previous round of review, we were asked to add information on the mean absolute errors (which is not on the figure). Despite we agree that there is some redundant information here, we also think that the reader can benefit from the quantitative information provided here, as quantitative values are rarely reported in the text of the manuscript.*

- Fig. 5 is very difficult to read.

*We agree that a lot of information is provided in the figure. However, it allows having an overall impression of the adjustment factors for the four regions, with the possibility to focus on specific glaciers or to detect outlier glaciers in the regions.*

- Fig. 6 illustrates well what is confusing me: why do the correction factors have trends? I think that Fig. 6 would also be a good opportunity to show actual data instead of correction factors, which is a very abstract notion for glaciologists…

*Thank you for this comment that we considered. However, we think that there are several reasons to prefer showing biases instead (factors between the winter mass balance and the models' estimates):*
- *The goal of the paper is to reduce the biases between the reanalysis and the winter mass balance data.*
- *Showing biases with factors allows comparing the results between different regions receiving different amounts of precipitation. Thus, we can understand in which regions the reanalyses are more biased.*
- *For a similar reason, the figure looks much cleaner by showing factors instead of the winter mass balance (or precipitation) data. In fact, there is a large variability between the winter mass balance data over the years (also because different glaciers are considered over the years depending on the data availability). In contrast, the factors between the winter mass balance data and the reanalysis precipitation (or GBR models) remains much more constant over the years.*

*Here, we report the two figures that confirm what we stated above:*

*Fig. 6 (new Fig. 5) in the manuscript (factors):*

[Figure]

*Figure with winter mass balance (or precipitation) data (in the figure above, we basically report the factor between the black line and the other lines (i.e. we report the biases)):*

[Figure]

*Regarding the first part of your comment, the biases (factors in this case) can have trends. This has been discussed in Sec. 5.2.1 (lines 425:428):*

In general, the year was included in the set of predictors for the GBRs because it may allow learning the climatological information and potential trends in terms of reanalysis biases against the $B_w$ data. This can be relevant in case reanalysis variables included in the predictors are not able to represent

the hypothetical trends of these biases, which can exist due to the increasing availability of observational data that could be assimilated by reanalysis models over the years.

- Fig. 6: when averaging factors, you should also plot the range (std dev) to show the robustness of the differences

*We agree, the standard deviation has been added to Fig. 6 (new Fig. 5).*

- L307: "confirming the importance of a specific optimization scheme depending on the goal of the model." -> I have to reiterate: what is the goal of the model?

*This has been made clear by following your previous suggestions.*

- L320: genuine question: is there any skill in ingesting data from very far away glaciers to interpolate in time?

*Thanks for your comment. We think that the answer is provided by Fig. 6 (b, d, f and h) in the new manuscript, and is discussed in Sec. 5.2.2.*

- L321: " In conclusion, filling data gaps is much simpler than estimating SWE on glaciers with no observations." yes, and this raises the question whether GBR is really needed for that or not (rhetorical question, requiring no change to the manuscript)

*We agree and we understand your comment. However, we think that it is fair to use the same kind of model (i.e. a GBR) for the interpretation of the differences between site- and season-independent models.*

- L325: see comment above: it is really unclear to me how the profiles are predicted…

*As for the comment above, we think that after the changes in the manuscript related to your comment above, this is now clear to the reader.*

- L363: " This suggests that complex models such as our GBRs are needed to adjust reanalysis to different glacier sites" -> this statement is made based on the benchmark model, which is not data informed. The paper does not say which model complexity is needed to achieve WMB reconstruction.

*Thank you. We modified the sentence by specifying what you correctly commented (line 378-380).*

- L380: "A disadvantage of tree-based algorithms, however, could be that this approach does not predict continuous values." -> I feel that this information should be shared much much earlier in the paper.

*Thank you. This information is now provided in the description of the model in the Methods section (3.2.2, lines 186:187).*

- Section 5.1.2 is highly speculative, short and not convincing.

*We agree. We added another reason that, intuitively, made us preferring a tree-based algorithm to other machine-learning methods, and that we think consolidates our choice. Here, we report the new section (lines 394:405):*

Intuitively, we preferred a tree-based algorithm given the high inhomogeneity in terms of spatial distribution of the considered glaciers. As further discussed in Sec. 5.2.1, a tree-based algorithm can exploit the coordinates (if provided as predictors) to easily split into individual sub-models adapted to different regions of the world (from the continental scale to the glacier-specific scale). Such operations would not be possible by using a simpler model such as a multiple linear regression. Also, it is less clear to us how artificial neural networks would behave given the considerable inhomogeneity of the spatial distribution.

In fact, we have chosen a tree-based algorithm because of its higher readability in terms of the predictors' usage compared to other machine-learning methods (e.g. Huysmans et al., 2011; Freitas, 2014). A disadvantage of tree-based algorithms, however, could be that this approach does not predict continuous values. Yet here, we aim at predicting an adjustment factor depending on a classification based on the used predictors, which is exactly the purpose of a tree-based algorithm. The choice of a gradient boosting instead of other tree-based algorithms (e.g. random forest (Breiman, 2001)) is motivated by the fact that gradient boosting is a gradient descent algorithm, where each additional tree tries to reduce the bias (which is the main goal of our study) rather than the variance of the predictions.

- In general, the discussion is by far the most interesting part of the paper. Many points related to how the method works or doesn't work are described here, and this is what makes the paper interesting.

*Thank you.*

- Section 5.2.4: I might be wrong, but I think that this is the only time the difference between the two reanalysis datasets is discussed? Does this justify the additional complexity of many of the figures? (I'm not suggesting changing the study design at this stage, but it is still a valid question).

*It is true that the differences are not discussed in depth and we put more emphasis on the methodology, but we think that the provided information can be useful for other scientists, e.g. for the calibration of models that need an adjustment of reanalysis precipitation. In fact, from the paper the reader can also have an idea on precipitation biases (of reanalysis datasets among the most used worldwide) for glaciers in different regions of the world. Also, we believe that showing that the model provides similar results with two different reanalyses, makes the study more robust.*

---

## Author Response (AR3)

**Response letter**

**Spatio-temporal reconstruction of winter glacier mass balance in the Alps, Scandinavia, Central Asia and Western Canada (1981-2019) using climate reanalyses and machine learning**

Guidicelli, M., Huss, M., Gabella, M., and Salzmann, N.

Dear Editor,

We would like to sincerely thank you, the two anonymous reviewers and the third reviewer, Fabien Maussion, for your constructive comments which improved the manuscript.

On behalf of all co-authors,

Matteo Guidicelli